# Evaluation and Prediction of Water Yield Services in Shaanxi Province, China

**Yanlin Li [1], Yi He [2,3,]*, Wanqing Liu [1,]*, Liping Jia [1] and Yaru Zhang [1]**

[1]  College of Urban and Environmental Sciences, Northwest University, Xi'an 710127, China
[2]  Shaanxi Key Laboratory of Earth Surface System and Environmental Carrying Capacity,
    Northwest University, Xi'an 710127, China
[3]  The Research Center of Soil and Water Conservation and Ecological Environment,
    Chinese Academy of Sciences and Ministry of Education, Yangling 712100, China
*   Correspondence: yihe@nwu.edu.cn (Y.H.); liuwqing@nwu.edu.cn (W.L.); Tel.: +86-29-8830-8342 (Y.H.)

**Abstract:** The water yield module of the InVEST model was used to estimate the water yield and its temporal and spatial variation characteristics in Shaanxi Province from 2000 to 2020. Moreover, the influences of future precipitation changes and land use changes on water yield in Shaanxi Province were discussed in the 2030s and 2050s. The results showed that: (1) from 2000 to 2020, the multi-year average water yields in northern Shaanxi, Guanzhong and southern Shaanxi were $33.23 \times 10^8$ m$^3$, $73.75 \times 10^8$ m$^3$, and $280.63 \times 10^8$ m$^3$, respectively; (2) the spatial pattern of water yield depth displayed a characteristic of gradually increasing from north to south; (3) under the precipitation change scenario, the water yield under different emission scenarios wa s in the order of RCP (Representative Concentration Pathways) 8.5 > RCP2.6 > RCP4.5; under the land use change scenario, the water yield depth of Shaanxi Province as a whole and in the three regions in the 2030s and 2050s showed a decline. The research results can provide scientific support for water ecological security, water resources, and regional high-quality sustainable development in Shaanxi Province.

**Keywords:** water yield; InVEST model; precipitation change; land use change; Shaanxi province

## 1. Introduction

With the continuous deepening of the construction of national ecological civilization, the importance of ecosystem services has become prominent. Paying attention to ecosystem services have become a hot issue [1,2]. Ecosystem services include all kinds of products and services provided by ecosystems directly or indirectly for human beings [3]. They are the basic guarantee for human survival and development. They are closely related to human well-being [4], and many studies related to ecosystem service assessments have been produced [5–7].

Water supply is one of the most important and valuable services of the ecosystem [8], which can meet the water needs of human irrigation, production, and life. It plays a crucial role in improving the environmental hydrological conditions of the basin and regulating the regional water cycle. Its changes can directly affect the climate, soil, vegetation, and hydrological conditions; it is an important indicator reflecting the services of the watershed ecosystem [9]. In recent years, the uncertainty of water supply caused by climate change has seriously threatened the security and stability of the ecosystem and affected the pace of the natural environment, human activities, and socioeconomic development [10]. Therefore, a quantitative evaluation of the water supply of the regional ecosystem is a necessary step for social development and progress.

Water yield is a quantitative expression of water supply services. The water yield service has strong temporal and spatial variability, and its change can directly affect the climate, hydrology, vegetation and soil conditions and is an important indicator of the

quality of the watershed ecosystem. However, water yield is affected by rainfall intensity, soil permeability, slope and vegetation [11]. The traditional water yield evaluation method is usually realized by using measured data, such as the soil water storage capacity method, the canopy interception surplus method, the precipitation storage method, the annual runoff method, and the underground runoff growth method [12]. With the continuous progress and upgrading of remote sensing and geographic information system (GIS) technology, hydrological models have played an essential role in the fields of ecology and hydrology [13–15]. More and more models can realize the simulation and evaluation of watershed water yield services, such as the Soil and Water Assessment Tool (SWAT) model [16–18], ARIES model [19,20] and the Integrated Valuation of Ecosystem Services and Trade-offs (InVEST) model [21–23].

The InVEST model water yield module assumes that the water yield of each grid cell is collected at the watershed outlet in the form of runoff, and the water supply is the water yield after subtracting the actual evapotranspiration from the precipitation of each grid cell. The model does not distinguish surface water, groundwater, and base flow. The water yield of the InVEST model comes from precipitation minus actual evapotranspiration. Aboveground, the actual evapotranspiration comes from the vegetation transpiration and surface evaporation, and the vegetation transpiration is related to the vegetation coefficient, seasonal constant and other parameters; underground and in the soil, the water is discharged through the root system and then further evaporated by the plant, which is reflected in the parameters such as the effective available water for vegetation, the available water for vegetation, and the root depth in the model.

Compared with other models, the application of the InVEST model is more in-depth and widespread all over the world [22]. The InVEST model has obvious advantages over other models in terms of data input, parameter calibration, spatial analysis function, result visibility, and being open source [13,24–26]. The InVEST model is a model system used for ecosystem services evaluation. Its water yield service evaluation is realized based on the Budyko equation, and fully considers the spatial differences in the soil permeability of different land use types and the influence of terrain and other factors on runoff [26]. The model can not only quantify the water yield in the study area, but can also express the spatial heterogeneity and temporal dynamics of water yield. The water yield module of the InVEST model has been widely used in many countries and regions and has achieved good application results [27–29]. The advantages of the InVEST model, the wide application of scholars, and the need for an assessment of the water yield services in Shaanxi Province have all played a practical role in promoting the research in this paper. Through the annual water yield module, this study mainly obtained the temporal and spatial changes in water yield in Shaanxi Province from 2000 to 2020 and the impact of future precipitation and land use changes on water yield changes.

In recent decades, the combined impact of climate change and human activities has led to global changes in river hydrological conditions [30,31]. Climate change affects the global water cycle model [32] to a certain extent and then affects water supply. As an intuitive reflection of human activities, land use change can affect the regional hydrological cycle process by changing water evaporation, infiltration, plant water conservation, and the available water in rivers and groundwater [33], and can then affect the supply capacity of the water supply services. Studying and predicting the temporal and spatial variations in water yield under land use and climate change can provide policy support for future socioeconomic development strategies and the sustainable development of the ecological environment. Through the use of data in the water yield module of the InVEST model and the analysis of existing studies, it has been demonstrated that water yield and climate factors are closely related to land use. Therefore, we can improve regional water yield by indirectly changing how land is used and intervening in factors affecting climate change. For example, we can increase regional water yield by increasing vegetation coverage, reducing bare land area, reducing the evapotranspiration, and improving the water conservation capacity [34].

Land use affects regional water yield by changing runoff and infiltration. The built up land is mainly composed of hardened pavement and buildings. Most of the surface water formed by precipitation will converge into runoff, while cropland, grassland and forest land are mainly composed of vegetation. Compared with built up land, there is a certain water interception capacity and the runoff formed by precipitation will be comparatively less. Through the transformation between different land use types, the water yield can be adjusted, which is beneficial to the stability of water resources and the balance of ecosystems.

Through the concept of "sponge city" proposed previously, built up land can increase the water conservation capacity of urban construction areas [35]. Among cropland, forest land, grassland and built up land, the water conservation capacity of built up land is the strongest, and the water infiltration can be changed through the transformation of cultivated land and grassland to built up land [36].

As an important inland province in China, Shaanxi Province includes two major river basins; the Yangtze River and the Yellow River. Its water systems are widely distributed, with a large latitude span, and the north, middle, and south show great spatial heterogeneity in terms of the geographical environment. With the construction and continuous promotion of a series of ecological construction projects, the regional ecological environment has been significantly improved. However, with the rapid development of urban construction, the contradiction of water resource utilization is still very serious. Therefore, quantitative research on the temporal and spatial changes in water yield in Shaanxi Province is of great significance for the rational allocation of regional water resources, the determination of ecological function protection areas, and regional sustainable development. The existing analysis of some administrative regions and watersheds [37,38] in Shaanxi Province has also played a positive role in its ecological construction and environmental protection. So far, there have been many studies on the local regions of Shaanxi Province; however, there is still a lack of research based on the whole of Shaanxi Province and from the perspective of comparative analysis of northern Shaanxi, Guanzhong, and southern Shaanxi. Internal comparative analysis is conducive to the efficient utilization and rational allocation of resources in the province, providing a fuller understanding of regional differences to adjust measures according to local conditions.

The main research objectives of this paper were (1) analyzing the temporal and spatial heterogeneity of water yield in different regions of Shaanxi Province; (2) using the Patch-generating Land Use Simulation (PLUS) model and the Statistical Downscaling Model (SDSM) to simulate future land use and precipitation changes; (3) predicting the change characteristics of water yield in Shaanxi Province in the future; and (4) discussing the contribution rate of precipitation change and land use change to water yield change in Shaanxi Province.

## 2. Materials and Methods

### 2.1. Study Area

Shaanxi Province is situated in the hinterland of China, in the middle reaches of the Yellow River, at 105°29′–111°15′ E and 31°42′–39°35′ N (Figure 1). The terrain of Shaanxi Province is high in the north and south and low in the middle, with plateaus, mountains, plains, and basins. Beishan and Qinling divide Shaanxi into three natural regions: northern Shaanxi, Guanzhong, and southern Shaanxi. The northern part of Shaanxi is the Loess Plateau, with an altitude of 900–1900 m, the central part is the Guanzhong Plain, with an altitude of 460–850 m, and the southern part is the Qinba Mountains, with an altitude of 1000–3000 m. The Qinling Mountains traverse Shaanxi from east to west with the Yellow River system to the north. The main tributaries from north to south are the Weihe River, Kuye River, Jinghe River, Wuding River, Beiluo River, Yanhe River, etc. Shaanxi Province spans three different climatic zones. The climate characteristics of Shaanxi Province are: warm and dry in spring, less precipitation, the temperature rises quickly and erratically, and the weather is windy; summer is rainy and hot; autumn is humid and cool, and the

temperature drops rapidly; winters are dry and cold, with low temperatures and little snow and rain. The Qinling Mountains represent the climatic dividing line between the north and the south of China. Southern Shaanxi has a northern subtropical climate, most of Guanzhong and northern Shaanxi have a warm temperate climate, and the Great Wall in northern Shaanxi has a mid-temperate climate. The province's annual average temperature is 9–16 °C, increasing from north to south and from west to east. The annual average temperature of northern Shaanxi is 7–12 °C, Guanzhong is 12–14 °C, and southern Shaanxi is 14–16 °C.

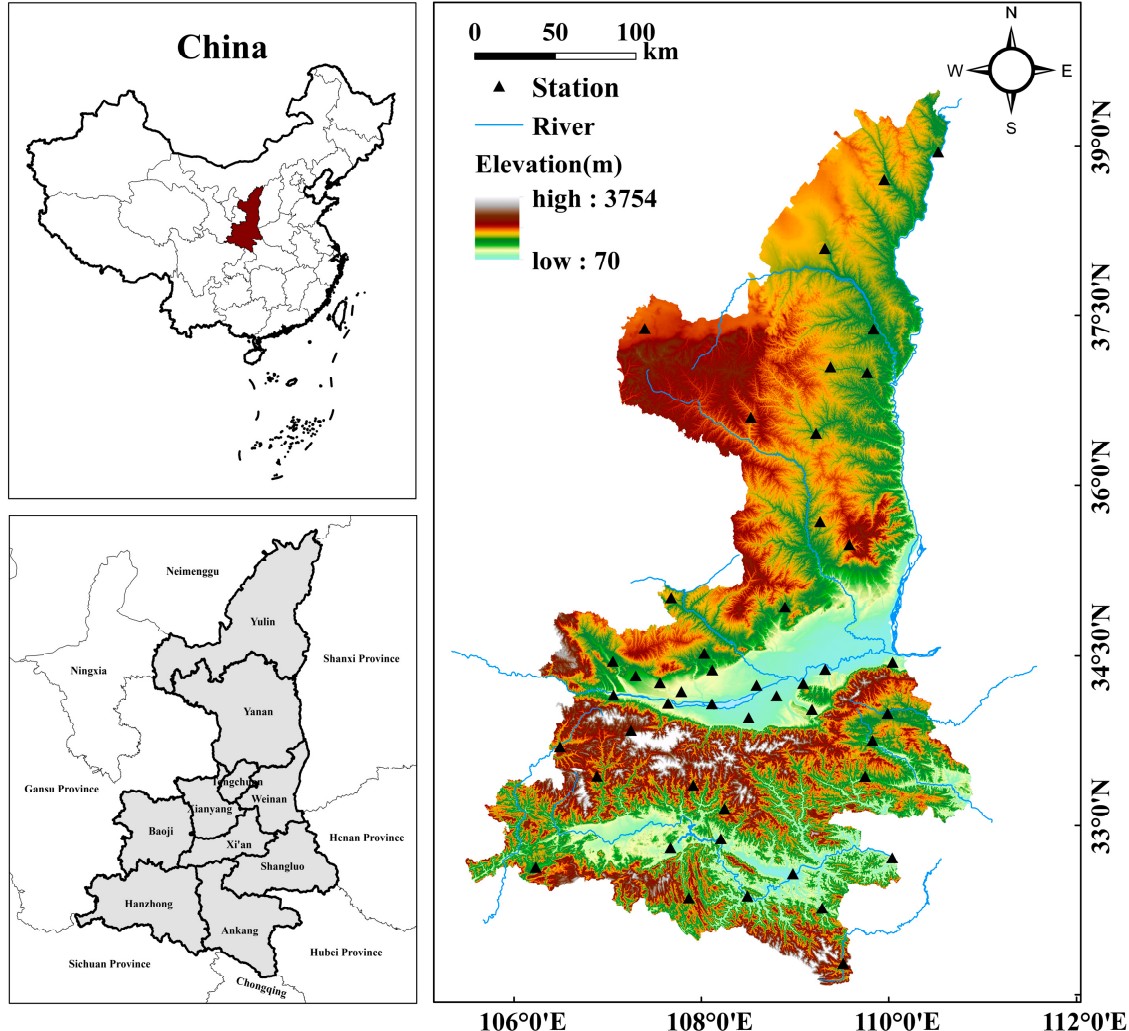

**Figure 1.** Stretch of the study area.

## 2.2. Overview of Soil in Shaanxi Province

The soil type in Shaanxi Province is shown in Figure 2. The main regional soils in Shaanxi Province are aeolian sandy soil, loess soil, paddy soil, tidal soil, newly deposited soil, marsh soil, saline alkali soil, etc. The zonal distribution of soil in Shaanxi Province is obvious. From the perspective of horizontal differentiation, the northern Shaanxi plateau is a chestnut soil and black loessial soil zone; Guanzhong Basin is a brown cinnamon soil zone; and the mountainous area in southern Shaanxi is a yellow-brown soil and yellow cinnamon soil zone. From the perspective of vertical differentiation, the Qinling Mountains and Daba Mountains are obvious. From the bottom to the top, the north slope of the Qinling Mountains is composed of cinnamon, brown, and dark brown soil that is subalpine meadow soil and primitive soil; the north slope of the Daba Mountain is composed of yellow cinnamon soil, yellow-brown soil, and brown soil from the bottom to the top.

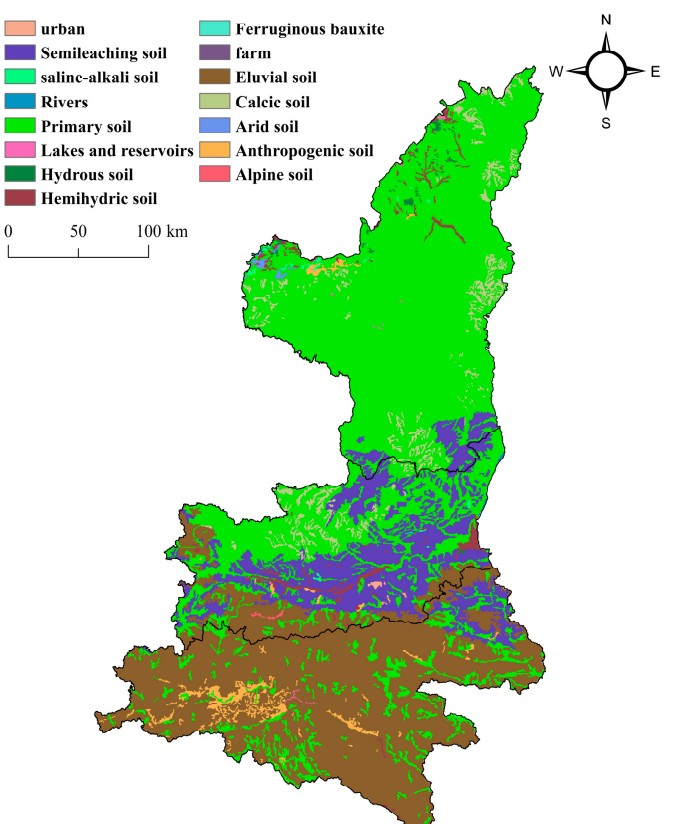

**Figure 2.** Spatial distribution of soil types in Shaanxi province.

*2.3. Research Methods*

2.3.1. Evaluation Method of the Water Yield Services

The water yield section of the InVEST model is based on the Budyko curve and annual evapotranspiration and precipitation [39]. According to the principle of the water cycle, water yield is calculated using parameters such as precipitation, plant transpiration, root depth, and soil depth. The specific calculation method is [25]:

$$Y_{jx} = \left(1 - \frac{AET_{xj}}{P_x}\right) \times P_x \tag{1}$$

where $Y_{jx}$ is the water yield of grid $x$ of the $j$th land use type (mm); $P_x$ is the annual average precipitation of grid $x$; and $AET_{xj}$ is the annual average evapotranspiration of grid $x$ of the $j$th land use type (mm). The InVEST model combines $AET$ and potential evapotranspiration ($PET$) using Equation (2), which was proposed by Budyko and later developed by Fu [40] and Zhang et al. [41].

$$\frac{AET_{xj}}{P_x} = 1 + \frac{PET_{xj}}{P_x} - [1 + (\frac{AET_{xj}{}^{\omega}}{P_x})]^{\frac{1}{\omega}} \tag{2}$$

$$PET_{xj} = k_c * ET_o \tag{3}$$

$$k_c = \begin{cases} \frac{LAI}{3}, & LAI \leq 3 \\ 1, & LAI > 3 \end{cases} \tag{4}$$

In Equation (3), $k_c$ is the vegetation coefficient, which is calculated from the vegetation leaf area index ($LAI$). In this paper, we use the improved Hargreaves equation [42] to calculate the potential evapotranspiration $ET_o$.

$$ET_o = 0.0013 \times 0.408 \times RA \times (T_{avg} + 17) \times (TD - 0.123P)^{0.76} \tag{5}$$

In Equation (5), $RA$ is the solar radiation at the top of the atmosphere (MJ m$^{-2}$ d$^{-1}$); $T_{avg}$ is the average value of the average daily maximum temperature and the average daily minimum temperature (°C); and $TD$ is the difference between the mean daily maximum temperature and the mean daily minimum temperature (°C).

$\omega_x$ is the ratio of the annual water availability to precipitation corrected for vegetation of grid $x$. In the InVEST model, the calculated expression given by Donohue et al. [43] was used to define $\omega_x$, as shown in Equation (6):

$$\omega_x = Z\frac{AWC_x}{P_x} + 1.25 \tag{6}$$

In Equation (6), $Z$ is a constant characterizing the seasonal characteristics of precipitation; $AWC_x$ is the effective available water for the vegetation of grid $x$, and the calculation formula is:

$$AWC_x = min(max\ Soil\ Depth_x, Root\ Depth_x) \times PWAC_x \tag{7}$$

In Equation (7), max $Soil\ Depth_x$ is the maximum soil depth of grid $x$; $Root\ Depth_x$ is the root depth of grid $x$; and $PWAC_x$ is the available water for the vegetation of grid $x$. The calculation formula is as follows:

$$PWAC_x = FMC - WC \tag{8}$$

In Equation (8), $FMC$ is the field water capacity and $WC$ is the wilting coefficient. They both need to use soil texture data for the calculation.

The water yield module in the InVEST model needs to calibrate the model by adjusting the seasonal factor $Z$ value when other parameters are determined. The seasonal factor $Z$ ranges from 1 to 30, representing regional precipitation distribution and other hydrogeological features [25].

### 2.3.2. SDSM Model

The SDSM model (Statistical Downscaling Model) is a downscaling tool established by Wilby et al. [44] and a combination method of multiple linear regression and a random weather generator [45], which overcomes the weakness that only the former will underestimate the interannual variability; at the same time, with the help of stochastic simulation tools, the variance of the daily series is closer to the observed value [46]. It is mainly used for the simulation of precipitation and temperature. The model considers the randomness of precipitation [47].

The working steps of the SDSM model are mainly composed of two parts: one is to establish the statistical relationship between the forecast and the predictors, and to determine the parameters required for the operation of the weather generator; the other is to use the GCM (Global Climate Model) data and the determined parameters to generate future daily series climate data [48,49]. Three climate change scenarios were selected in this study, namely RCP2.6, RCP4.5, and RCP8.5. RCP is the abbreviation of "Representative Concentration Pathways", which measures the concentration of greenhouse gases in the atmosphere. RCP2.6 represents the path with the lowest greenhouse gas emissions, followed by RCP4.5, and RCP8.5 represents the path with the highest greenhouse gas emissions.

In order to have higher confidence in the forecast results, the relationship between forecast quantities and predictors must first be established. Taking the daily precipitation data from the existing station data in the study area as the forecast, 26 large-scale predictors were selected. The predictors were derived from the prediction factors of historical and future climate change scenarios in the large-scale grid, corresponding to Shaanxi meteorological stations in the CanEsm2 dataset developed by the Canadian Environmental and Climate Change Modeling and Analysis Center in CMIP5. Through the analysis of the explained variance and partial correlation coefficient between the daily precipitation data of Shaanxi meteorological stations and 26 predictors, 3–5 prediction factors with the highest

correlation with precipitation were obtained, which can be used to predict precipitation in future climate change scenarios [50].

Due to the different completeness of the daily precipitation data of different meteorological stations, we selected 46 stations (Figure 1) with complete daily precipitation data from 1961 to 2005 from all meteorological stations in Shaanxi Province, including 11 stations in northern Shaanxi, 20 stations in Guanzhong and 15 stations in southern Shaanxi. Due to the distribution of the central plain in the Guanzhong area, there were many meteorological stations with high data integrity, so there were many stations available for selection.

To ensure the consistency with the time series length of the GCM data, the measured meteorological data from 1961 to 2005 were selected for the downscaling simulation to assess the credibility of the model; the period from 1961 to 1990 was set as the regular period, and the period from 1991 to 2005 was set as the verification period. The IPCC recommends base periods longer than 30 years, so this study selected 30 years to describe the climate characteristics of the region.

2.3.3. PLUS Model

The PLUS model is a patch-generated model for simulating land use changes based on raster data. Compared with the existing land use simulation models such as CLUE-S and CA-Markov, the PLUS model applies a new land use expansion analysis strategy (LEAS), which can better explore the reasons for changes in various land use types. In addition, the random forest method was used to find various land use expansion factors and driving forces to obtain the development probability of various land uses and the contribution rate of driving factors to the expansion of various land uses. This strategy combines the advantages of existing transformation analysis strategy and pattern analysis strategy, avoids analyzing the transformation types that grow exponentially with the number of categories, and retains the model's ability to analyze the mechanism of land use change, so that the results have better interpretability. The PLUS model contains a new multi-type seed growth mechanism based on the CA model of multi-type random patch seeds (CARS), and combines random seed generation and threshold reduction mechanisms. The model can simulate the automatically generated spatiotemporal dynamic simulation patches under the constraint of development probability, and can better simulate the changes in multi-type land use patches [51,52].

The simulation of land use in Shaanxi Province was divided into two steps: (1) Using the land use data from 2010 and 2020 as the benchmark data, the land use data in 2030 and 2050 were simulated, respectively, and the actual data in 2020 were compared with the simulated data. The Kappa coefficient was used as the judgment basis for the accuracy analysis. The Kappa coefficient was higher than 0.7, indicating that the simulation results were highly consistent with the actual data. (2) On the premise that the simulation accuracy met the requirements, the land use data simulation in 2030 and 2050 was carried out using the Markov-chain simulation.

Markov method: the Markov method is an analytical method that uses the current situation and trend in a variable to predict the future state and trend. The mutual transformation of different land use types in the region is a complex process, which is difficult to accurately describe with a functional relationship; that is, under certain conditions, the change in land use type conforms to the nature of the Markov random process, so it is feasible to use it to study the dynamic transformation of land use types, and the mutual transformation between land types can be quantitatively obtained.

*2.4. Precipitation Change and Land Use Scenario Analysis*

Precipitation and land use are the most important factors affecting the water yield module of the InVEST model. This paper used scenario analysis and difference comparison methods to explore the relative contribution of future precipitation changes and land use changes to water yield changes in Shaanxi Province. Since the land use simulation did not distinguish between carbon emission scenarios, the comparison of the climate change

scenario and land use change scenario referred to the water yield data under moderate carbon emissions.

The quantification of the contribution rate was achieved by simulating the process of water yield and precipitation change and land use change scenarios. The calculation method is as follows [53]:

$$C_p = \frac{\Delta p}{\Delta_p + \Delta_l} \times 100\% \tag{9}$$

$$C_l = \frac{\Delta l}{\Delta_p + \Delta_l} \times 100\% \tag{10}$$

where is the contribution rate of precipitation changes to water yield changes; $C_l$ is the contribution rate of land use changes to water yield changes; and $\Delta_p$ and $\Delta_l$ are the amount of water yield changes under precipitation change scenarios and land use change scenarios, respectively.

Precipitation change scenario: we kept the land use data input by the water yield module unchanged (based on the land use data in 2020) and, using the precipitation changes, we input the precipitation in the 2030s and 2050s in sequence. The difference between the simulated water yield under this scenario and the simulated water yield with simultaneous changes at the same time in terms of precipitation and land use was the impact of precipitation changes on water yield changes. In this way, the impact of climate change on water yield was discussed if the underlying surface did not change.

Land use change scenario: we kept the precipitation data input by the water yield module unchanged (based on the average precipitation data from 2000 to 2020) and, using the land use changes, we input the land use data in the 2030s and 2050s in turn. The difference between the simulated water yield under this scenario and the simulated water yield with simultaneous changes in precipitation and land use together was the influence of land use changes on water yield changes. Then, the effect of the underlying surface change on water yield was discussed if the climate change did not occur.

### 2.5. Data Sources and Preparation

The data required by the water yield module of the InVEST model were precipitation, reference evapotranspiration, land use, depth to root restriction, plant available water fraction, biophysical table, and the seasonality factor Z. In this paper, the geographic coordinate system of the Shaanxi raster data used was CGCS 2000; the spatial resolution was 1km. The table of biophysical coefficients is shown in Table 1, in which LULC_desc represents the name of the land use type; LULC_veg represents the land use type, where vegetation cover was assigned a value of 1 and other land use types were assigned a value of 0; and Kc represents the plant evapotranspiration coefficient, determined according to Sharp et al. [25] and Bao et al. [54]. Other related data sources and processing methods are shown in Table 2.

**Table 1.** Biophysical table for water yield.

| Lucode | LULC_desc | LULC_veg | Root_depth | Kc |
|--------|-----------|----------|------------|------|
| 1 | Cropland | 1 | 300 | 0.65 |
| 2 | Forest | 1 | 5000 | 1 |
| 3 | Grassland | 1 | 500 | 0.65 |
| 4 | Water area | 0 | 1 | 1 |
| 5 | Built up area | 0 | 1 | 0.3 |
| 6 | Unused land | 0 | 1 | 0.5 |

**Table 2.** Data sources and parameters settings.

| Data | Data Sources/Parameters Settings |
|---|---|
| Precipitation, Reference evapotranspiration, Temperature | China National Earth System Science Data Center (http://www.geodata.cn/data accessed on 3 May 2022) |
| Depth to root restriction layer | China National Tibetan Plateau Data Center (http://www.tpdc.ac.cn/zh-hans/data/ accessed on 5 May 2022) |
| Soil data, Land use, Watershed, Socioeconomic data | Resource and Environment Science and Data Center, Chinese Academy of Sciences (http://www.resdc.cn/ accessed on 6 May 2022) |
| Plant available water fraction | PAWC = FMC − WC, FMC is the field water capacity, and WC is the wilting coefficient; both are calculated from soil texture data. |
| Z parameter | Z is the seasonal constant, which was obtained by repeated verification according to the recorded water resources in the Shaanxi Provincial Water Resources Bulletin and simulated water yield, where $Z_1$ = 15.7, $Z_2$ = 14.29, and $Z_3$ = 3.55 |
| Digital Elevation Model (DEM) | Geospatial Data Cloud, Chinese Academy of Sciences (https://www.gscloud.cn/ accessed on 6 May 2022) |
| The road data | OpenStreetMap (https://www.openstreetmap.org/ accessed on 8 May 2022.). |

Notes: $Z_1$, $Z_2$, and $Z_3$ represent the Zhang coefficient in northern Shaanxi, Guanzhong, and southern Shaanxi, respectively.

*2.6. Research Ideas*

The evaluation and prediction of water yield services in Shaanxi Province mainly include the evaluation of water yield services in Shaanxi Province from 2000 to 2020, and the impact of future precipitation and land use on water yield services in Shaanxi Province. In the discussion, the relationship between different parameters and water yield as well as the calibration and limitations of water yield assessment were discussed. The following is the research flowchart of this paper (Figure 3).

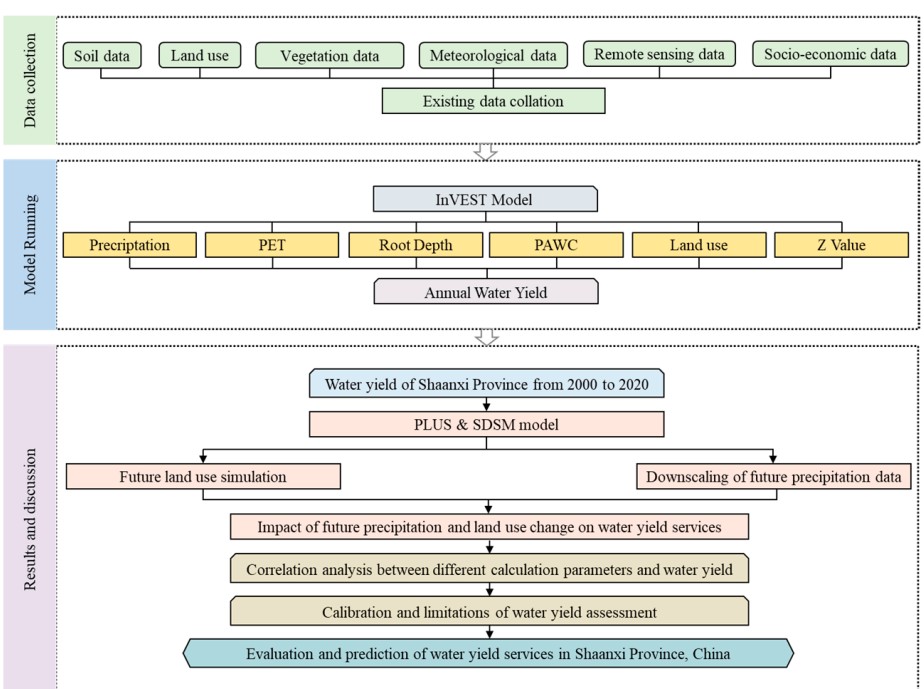

**Figure 3.** Research flowchart of this paper.

## 3. Results and Discussion

*3.1. Characteristics of Interannual Variation in Water Yield*

The interannual changes in actual evapotranspiration, potential evapotranspiration, precipitation, and water yield depth in Shaanxi Province from 2000 to 2020 are shown in Figure 4. The multi-year average potential evapotranspiration was 1056.43 mm, and the multi-year average actual evapotranspiration was 470.89 mm, and the changes between the two were relatively stable. The annual precipitation was 659.89 mm, the highest precipitation was 860.39 mm (2003), and the lowest precipitation was 566.99 mm (2001). Compared

with evapotranspiration, the annual precipitation showed a more considerable fluctuation, showing an insignificant upward trend. For the interannual changes in actual evapotranspiration and precipitation, it could be seen that the changes in actual evapotranspiration were highly correlated with the changes of precipitation, which meant that most of the precipitation through actual evapotranspiration was due to the atmosphere again. Compared with actual evapotranspiration and potential evapotranspiration, precipitation has the strongest correlation with water yield. In the InVEST model, the water yield follows the principle of water balance. The regional water yield is precipitation minus actual evapotranspiration, but the actual evapotranspiration is also affected by precipitation, so precipitation plays a major role in the water yield. Under the influence of these meteorological elements, the average water yield depth in the study area for many years was 189 mm and the water yield was $387.61 \times 10^8$ m$^3$, showing an insignificant upward trend overall. The highest water yield depth in Shaanxi Province was 356.15 mm (2003) and the water yield was $710.77 \times 10^8$ m$^3$, and the lowest water yield depth was 114.54 mm (2001) and the water yield was $234.53 \times 10^8$ m$^3$; this was consistent with the distribution time of high and low values of precipitation.

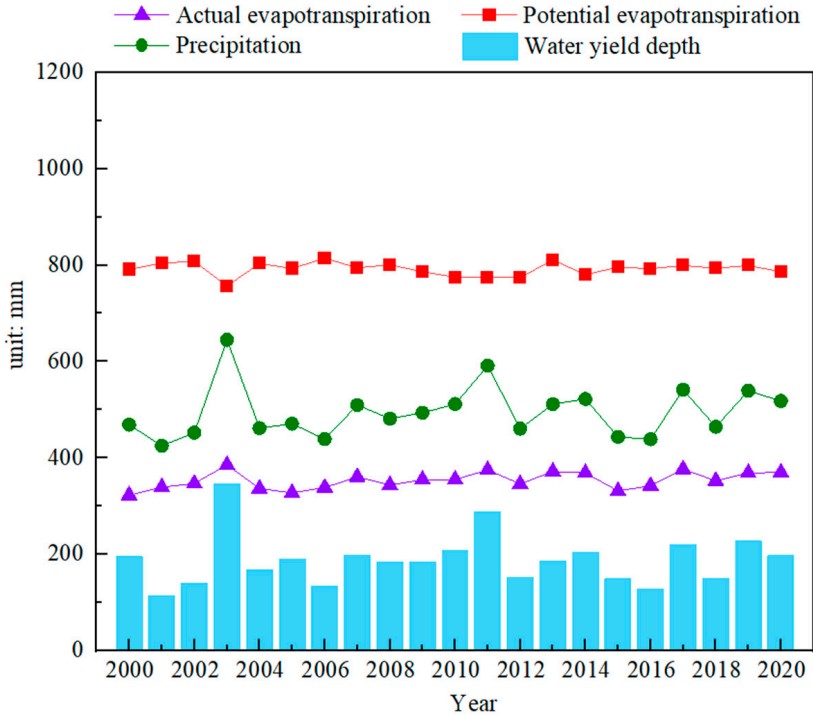

**Figure 4.** Actual evapotranspiration, potential evapotranspiration, precipitation, and water yield depth in the Shaanxi Province from 2000 to 2020.

The changes in water yield depth in the three major regions of Shaanxi Province from 2000 to 2020 are shown in Figure 5. The multi-year water yield depth in northern Shaanxi was 41.39 mm, and the multi-year water yield was $33.23 \times 10^8$ m$^3$, indicating an upward trend with a rising rate of 7.5 mm/10 a; the multi-year water yield depth in Guanzhong was 133.16 mm, and the multi-year water yield was $73.75 \times 10^8$ m$^3$, showing the same trend as in northern Shaanxi, with a rising rate of 10.9 mm/10 a; the annual water yield depth in southern Shaanxi was 401.31 mm, and the annual water yield was $280.63 \times 10^8$ m$^3$, revealing a certain degree of decline, with a reduction rate of 14.2 mm/10 a. Among the three major regions, the water yield in southern Shaanxi ranked highest, followed by Guanzhong and northern Shaanxi, indicating that water yield was closely related to meteorological elements such as precipitation, natural environmental elements, and latitude position.

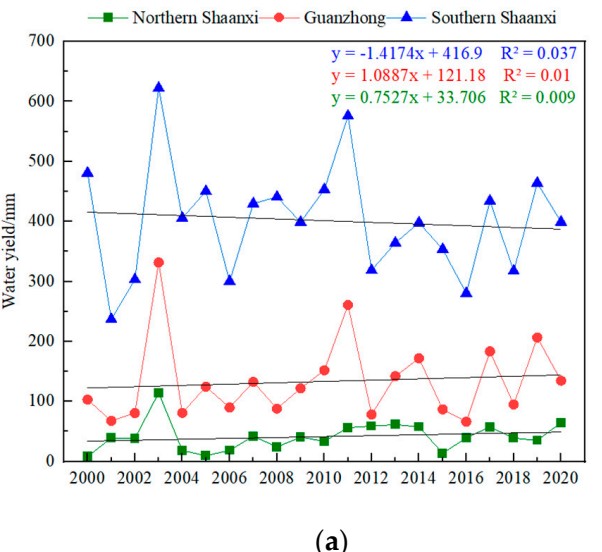
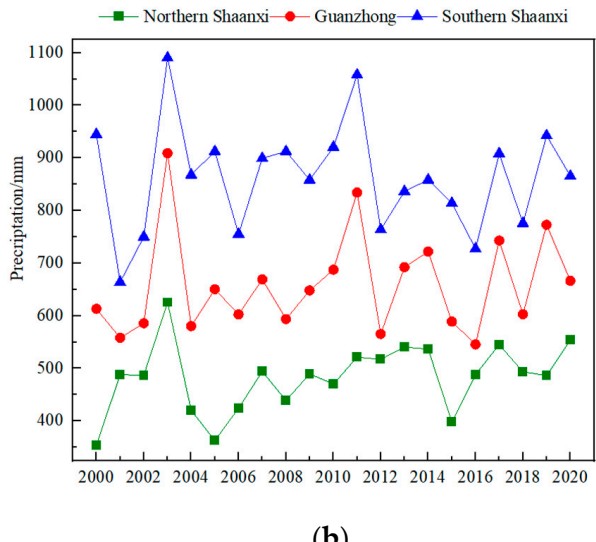

<div style="text-align:center">(<b>a</b>)   (<b>b</b>)</div>

**Figure 5.** Water yield (**a**) and Precriptation (**b**) in northern Shaanxi, Guanzhong, and southern Shaanxi.

### 3.2. Spatial Distribution of Precipitation and Actual Evapotranspiration from 2000 to 2020

The spatial distribution of precipitation and actual evapotranspiration in Shaanxi Province from 2000 to 2020 is shown in Figure 6. The precipitation data of Shaanxi Province from 2000 to 2020 were cut and synthesized with the national geoscience data center's 1901–2021 1 km resolution monthly precipitation data set in China.

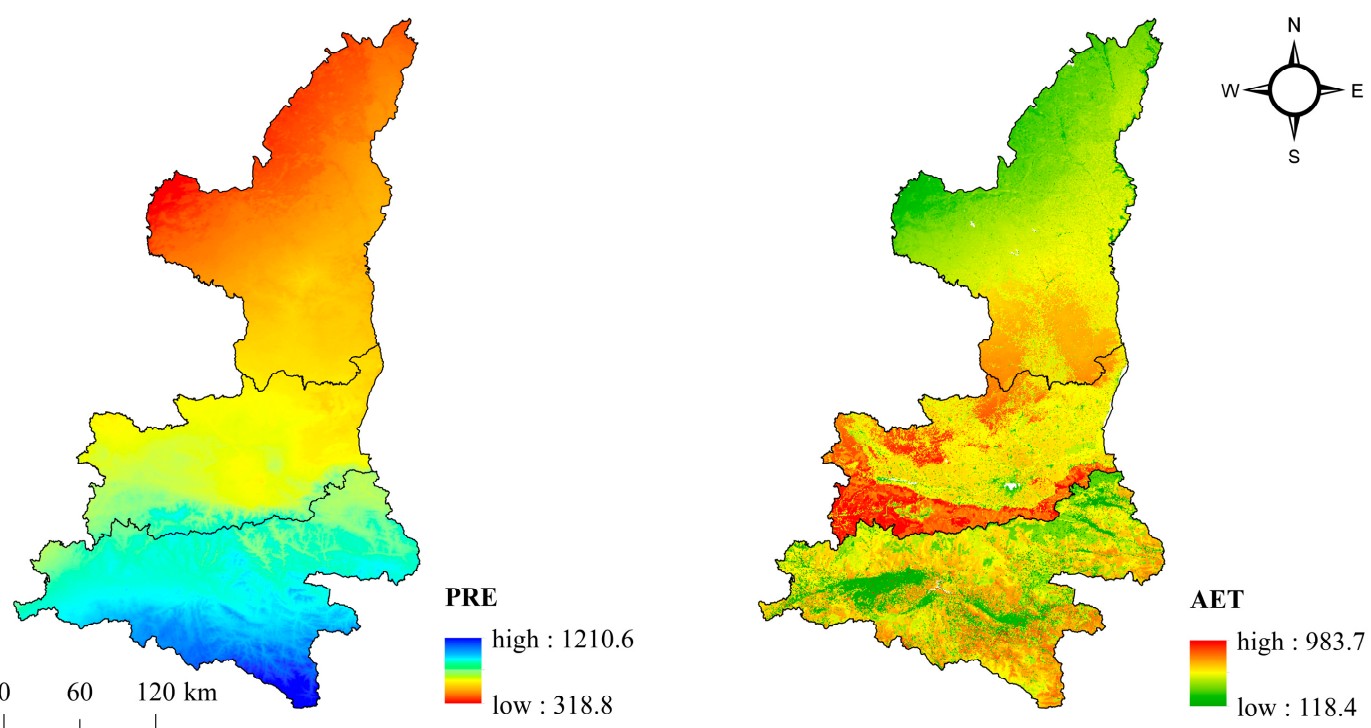

**Figure 6.** The spatial distribution of precipitation and actual evapotranspiration in Shaanxi Province from 2000 to 2020.

The precipitation in Shaanxi Province has risen slowly at the rate of 17.48 mm/10 a in over 20 years. From 2000 to 2020, the highest value of precipitation in Shaanxi Province occurred in 2003, at 860.39 mm, and the lowest value occurred in 2001, at 567 mm. The multi-year average precipitation was 659.89 mm. The years above the average value

accounted for 42.86% of the total, and the years below the average value accounted for 57.14%. The overall fluctuation in precipitation was large, and the annual distribution of precipitation was uneven.

The precipitation not only has great interannual variation, but also has great spatial difference between the north and south. The precipitation of Shaanxi Province shows a spatial pattern of a gradual increase from north to south. The average annual precipitation in northern Shaanxi is 482.53 mm, that in Guanzhong is 658.45 mm, and that in southern Shaanxi is 863 mm. Approximately 43.08% of the precipitation is concentrated in southern Shaanxi, with a maximum of 861.36 mm in Hanzhong City and 931.65 mm in Ankang City, which belongs to the Hanjiang River basin.

The actual evapotranspiration data of Shaanxi Province from 2000 to 2020 were the simulation results of the water yield of the InVEST model. The results show that the average annual actual evapotranspiration in Shaanxi Province was 470.89 mm, including 441.49 mm in northern Shaanxi, 525.3 mm in Guanzhong and 461.91 mm in southern Shaanxi. The spatial distribution of actual evapotranspiration was highly consistent with that of the vegetation. The actual evapotranspiration in northern Shaanxi, the Guanzhong Plain and residential areas in southern Shaanxi was low. The actual evapotranspiration in Laoshan Mountain, Ziwu Mountain and Huanglong Mountain in northern Shaanxi, the northern foot of Qinling Mountain in central Shaanxi, and Daba Mountain in southern Shaanxi were high.

### 3.3. Spatial Distribution of Water Yield from 2000 to 2020

The spatial distribution of water yield depth in Shaanxi Province is presented in Figure 7. The depths of water yield in Shaanxi Province in 2000–2005, 2006–2010, 2011–2015, and 2016–2020 were 192.59 mm, 182.13 mm, 196.53 mm, and 184.76 mm, respectively, and the water yields were $395.97 \times 10^8$ m$^3$, $374.46 \times 10^8$ m$^3$, $404.07 \times 10^8$ m$^3$, and $379.87 \times 10^8$ m$^3$, respectively. The water yield in Shaanxi Province showed apparent spatial and temporal heterogeneity and exhibited a gradually increasing trend from north to south in space.

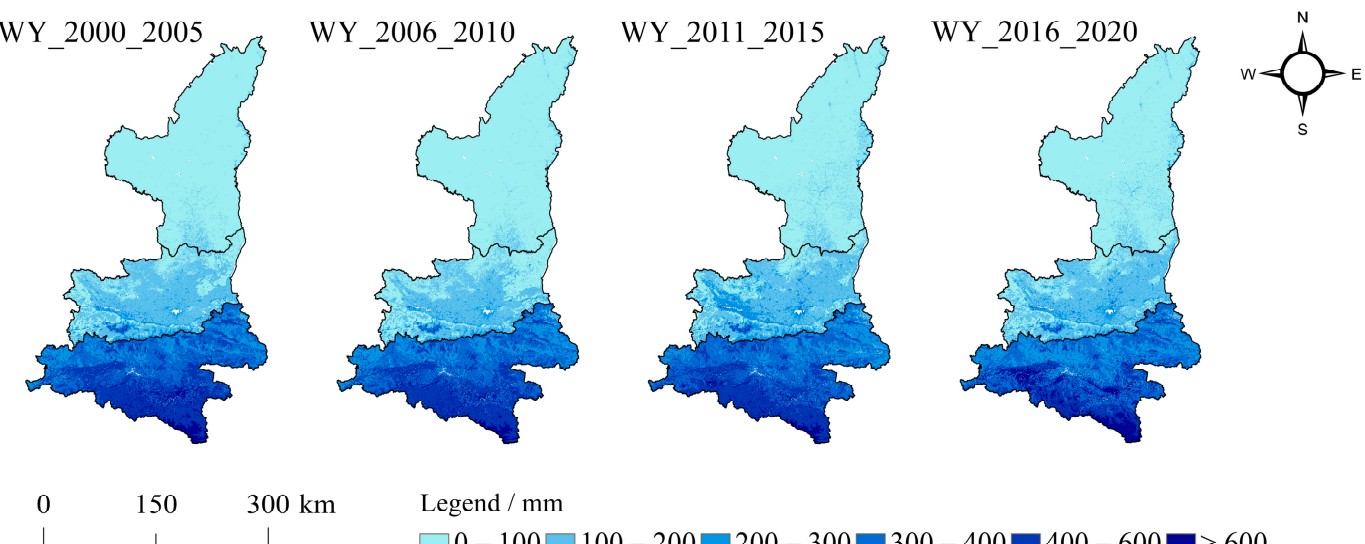

**Figure 7.** Spatial distribution of water yield depth in Shaanxi Province in 20002005, 20062010, 20112015, and 20162020.

The distribution maps of water yield depth in northern Shaanxi, Guanzhong, and southern Shaanxi are shown in Figure 8. More than 90% of the water yield depths in northern Shaanxi were 0–100 mm, and a few were in the range of 100–200mm. In general, it was larger in the south than in the north, and the south was more sensitive to various external factors, showing a greater degree of fluctuation in different periods. The southern

water yield increased significantly from 2010 to 2014. Approximately 30% of the water yield depth of Guanzhong was 0–100 mm, and 58.03% was 100–200 mm. High-value regions were mainly concentrated in the west of Guanzhong, and low-value regions were located east of Guanzhong. Baoji City and Xi'an City in the southwestern part of the city had a maximum distribution of water yield due to the influence of the Wei River. From the time scale, the water yield in the northwest of Guanzhong showed noticeable changes over time. The water yield depth and water yield in southern Shaanxi were the highest in the province, with 94.39% having a water yield depth of 200–600 mm. The spatial distribution of water yield in southern Shaanxi was relatively uniform. Due to the influence of the Qinling Mountains blocking the warm and humid airflow from the south, there was sufficient water vapor. Due to the impact of the Han River and Jialing River, tributaries of the Yangtze River, the surface and groundwater resources were adequate; as a result, the water yield in southern Shaanxi was relatively large.

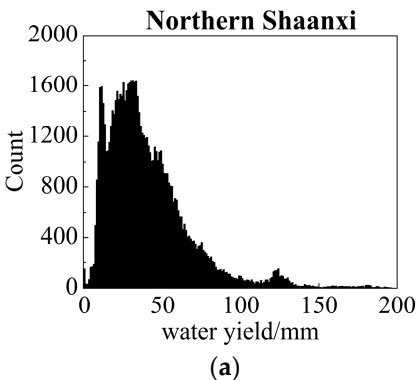 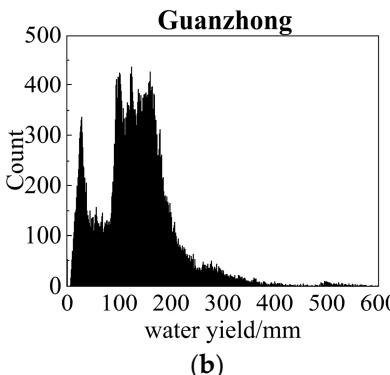 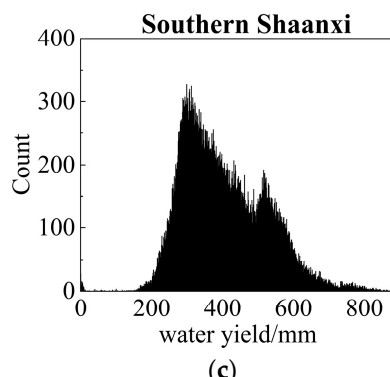

**Figure 8.** Distribution of water yield depth in Northern Shaanxi (**a**), Guanzhong (**b**), and Southern Shaanxi (**c**).

### 3.4. Precipitation and Land Use Simulation Results

3.4.1. Simulation Results of Precipitation Based on the SDSM Model

Using the SDSM model to downscale precipitation data, it was first necessary to clarify the statistical relationship between precipitation and related predictors and to determine the parameters required for the operation of the weather generator. Table 3 shows the selection results of predictors for each meteorological station in Shaanxi Province. The established statistical relationship was applied to the NCEP reanalysis data from 1991 to 2005 to generate the daily series of precipitation in this period, and the degree of agreement between the simulated precipitation and the measured precipitation was compared and analyzed. In this paper, the Nash efficiency coefficient (NS) was used to judge the simulation effect; the NS of 20% of the stations was lower than 0.65, meaning the simulation effect was poor, and the NS of 80% of the stations exceeded 0.65, meaning the simulation effect was good; the specific NS coefficient values can be found in Figure 9. Zichang Station and Pingli Station, located in northern and southern Shaanxi, respectively, had the worst simulation effect, with their NS being lower than 0.5. Shenmu, Zhidan, Lintong, Luochuan, and Xi'an Stations had the best simulation effect, with their NS greater than 0.93. The Shenmu, Zhidan and Luochuan Stations are located in northern Shaanxi, and the Lintong and Xi'an Stations are located in Guanzhong. There was no obvious spatial correlation between good and bad sites.

After completing the calibration of the model, for the 2030s, 2050s and three scenarios of RCP2.6, RCP4.5, and RCP8.5, the future scenario sequences output from the GCM were input into the SDSM model to obtain the station's future climate sequence data. Due to the randomness of precipitation, the simulation effect of some stations was poor, so it was necessary to correct the error of the precipitation simulation result using the scale factor [55,56], and the results are shown in Figure 10.

**Table 3.** Selection results of predictors for each meteorological station.

| Station | Predictors-PRE | Station | Predictors-PRE |
| --- | --- | --- | --- |
| Fugu | Mslp, p500, p8_z | Xi'an | mslp, p1_v, p500, prcp, s500 |
| Yulin | mslp, p500, p8_z | Lintong | mslp, p500, prcp, s500 |
| Shenmu | mslp, p500, s500 | Weinan | p1_v, p500, prcp, s500 |
| Dingbian | mslp, p1_v, p500 | Lantian | mslp, p500, p8_v, s500 |
| Zichang | mslp, p1_v, p500 | Xianyang | mslp, p1_u, p500, prcp, s500 |
| Suide | mslp, p5_v, p500, s500 | Tongguan | mslp, p1_v, p8_z, prcp |
| Qingjian | mslp, p1_v, p500, s500 | Luonan | mslp, p1_v, p500, prcp |
| Zhidan | mslp, p500, prcp, s500 | Fengxian | mslp, p1_v, p500, p8_v, prcp |
| Yanan | mslp, p500, prcp, s500 | Liuba | mslp, p1_v, p500, prcp |
| Changwu | mslp, p1_v, p500, prcp | Xixiang | mslp, p1_v, p500, p8_z |
| Luochuan | mslp, p500, p8_z, s500 | Huxian | mslp, p1_v, p500, prcp, s500 |
| Huanglong | mslp, p1_v, p500, p8_z | Foping | mslp, p1_v, p500, prcp |
| Tongchuan | mslp, p1_v, p500 | Ningshan | p1zh, p500, p8_z |
| Baoji | mslp, p5_v, p500, p8_v, prcp | Shangzhou | mslp, p1_v, p500, prcp |
| Qianyang | mslp, p5_v, p500, prcp | Shanyang | mslp, p1_v, p500, prcp |
| Qishan | mslp, p5_v, p500, prcp | Ningqiang | mslp, p500, p8_z, s500 |
| Fengxiang | mslp, p5_v, p500, prcp | Ziyang | mslp, p1_v, p500, p8_z, s500 |
| Fufeng | mslp, p1_v, p500, prcp | Shiquan | mslp, p1_v, p500, p8_z |
| Meixian | mslp, p5_v, p500, prcp | Zhenba | p1_v, p500, p8_v, p8_z |
| Taibai | mslp, p5_v, p500, p8_v, prcp | Ankang | mslp, p500, p8_z |
| Yongshou | mslp, p5_v, p500, p8_v, prcp | Pingli | mslp, p1_v, p500, p8_z |
| Wugong | p5_v, p500, p8_v, prcp | Baihe | mslp, p500, p8_z |
| Qianxian | mslp, p5_v, p500, p8_v, prcp | Zhenping | mslp, p1_v, p500, p8_z |

Notes: mslp represents sea level pressure; p1_, p5_, and p8_ represent sea level pressure, 500 hPa, and 850 hPa, respectively; z represents vorticity; u represents zonal wind speed; v represents radial wind speed; zh represents divergence; p500 represents geopotential at 500 hPa height; s500 represents specific humidity at 500 hPa; and temp represents near-surface temperature.

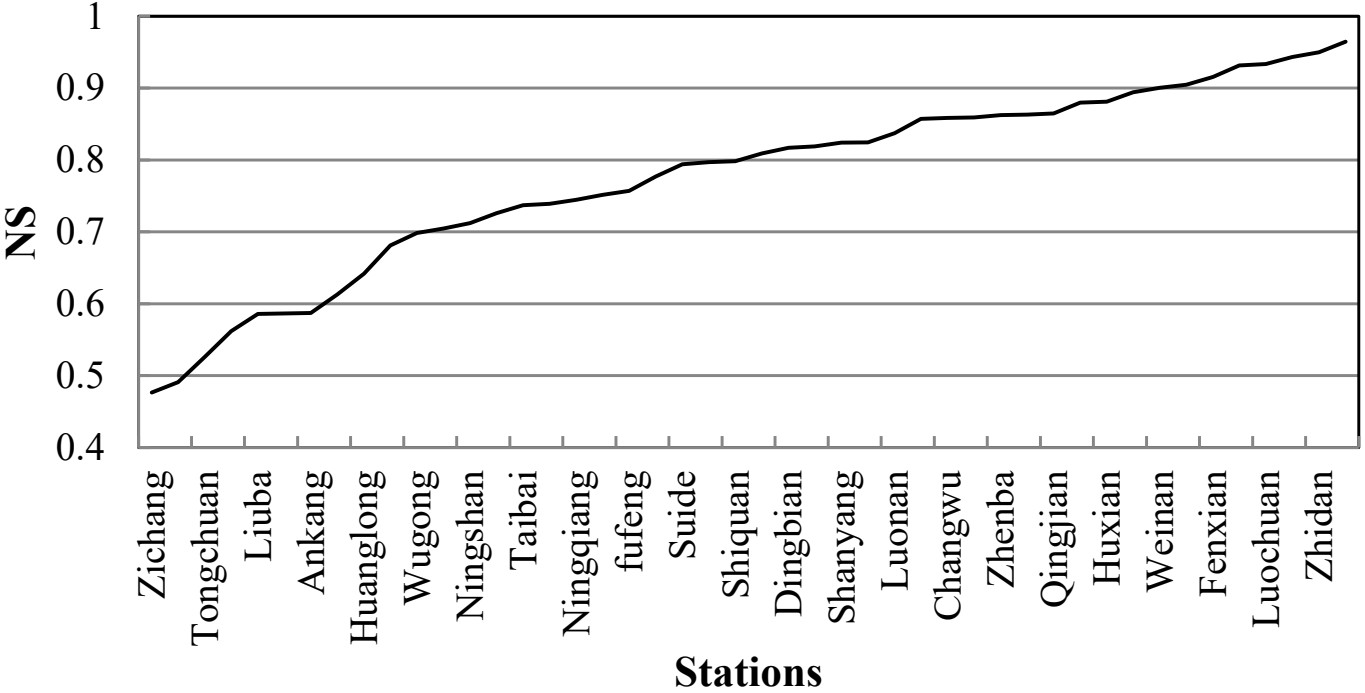

**Figure 9.** Simulation results of precipitation data at each station.

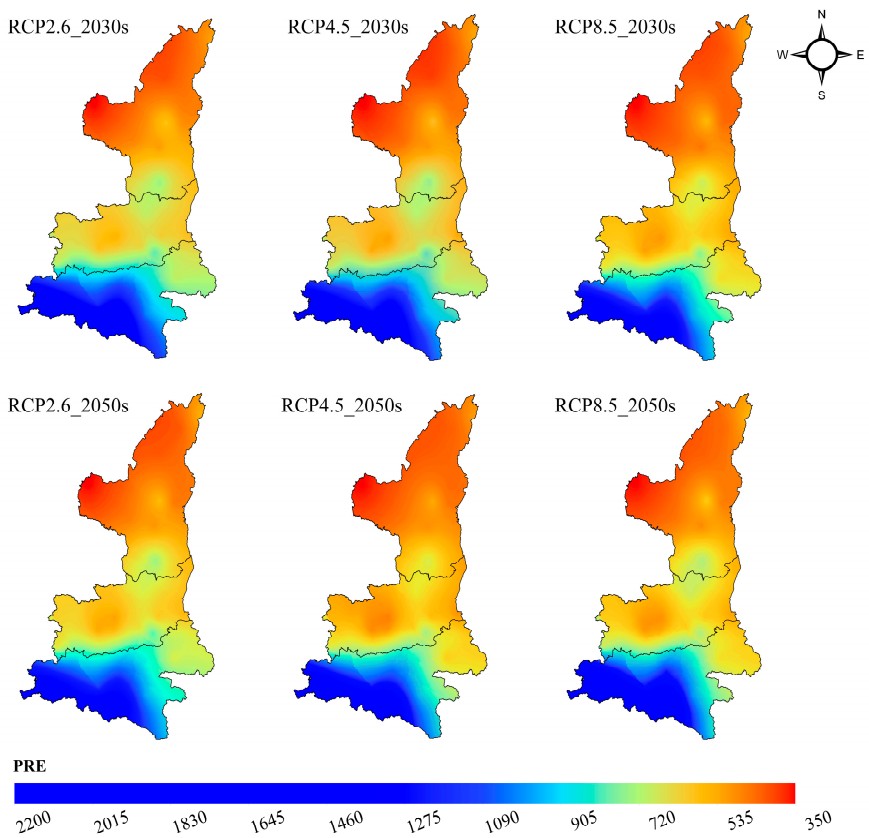

**Figure 10.** Precipitation in the 2030s and 2050s under RCP2.6, RCP4.5, and RCP8.5 scenarios.

The results showed that the precipitation in Shaanxi Province would increase from south to north in space in the future. The temperature obtained from the 2030s and 2050s downscaling would be higher than that in 2020 under different carbon emission models. In the 2030s, under the scenarios of RCP2.6, RCP4.5, and RCP8.5, the precipitation would be 742.49 mm, 739.98 mm, and 763.64 mm with an increase of 7.49%, 7.13% and 10.56% compared with 2020, respectively; under the scenarios of RCP2.6, RCP4.5 and RCP8.5 in the 2050s, the precipitation would be 769.7 mm, 767.95 mm and 813.59 mm, which is 11.43%, 11.18% and 17.79% higher than that in 2020, and 3.66%, 3.78% and 6.54% higher than that in the 2030s, respectively. Under the RCP4.5 scenario, the precipitation in Shaanxi Province would be the lowest in the 2030s and 2050s, while under the RCP8.5 scenario the precipitation in Shaanxi Province would be the highest in the 2030s and 2050s. From the comparison results of precipitation under three carbon emission scenarios, it can be concluded that, under the medium carbon emission scenario, the precipitation would be the lowest, followed by the RCP2.6 and RCP8.5 scenarios.

### 3.4.2. Land Use Simulation Results Based on the PLUS Model

Using the PLUS model for land use simulation first required conversion of the format of the land use data, converting the land use data in northern Shaanxi, Guanzhong, and southern Shaanxi in 2010 and 2020 from the "tif" format to "uc" format, and then extracting the land use expansion data from 2010 to 2020. Next, we entered the land use expansion data and all the contribution factors files that were prepared in advance into the LEAS module to obtain various land use development probabilities. Then, we input the 2010 land use data, various land use development probability data, restricted development area data, and related parameters into the CARS module to obtain the 2020 land use simulation data. The 2020 land use simulation data and actual land use data were used to verify the simulation effect. The results are shown in Table 4. The Kappa coefficient of the three regions exceeded 0.7; therefore, the simulation effect was excellent.

**Table 4.** Kappa coefficient of land use simulation results in different regions of Shaanxi Province in 2020.

| Area | Northern Shaanxi | Guanzhong | Southern Shaanxi |
|---|---|---|---|
| Kappa coefficient | 0.74 | 0.86 | 0.78 |

We then used a Markov chain to predict the land use in the 2030s and 2050s, and obtained the number of various types of patches; then, we inputted the actual land use data in 2020, various land use development probability data, and various types of parameter data into the CARS module again to obtain the spatial data of land use in Shaanxi Province in the 2030s and 2050s. The results are exhibited in Figure 11.

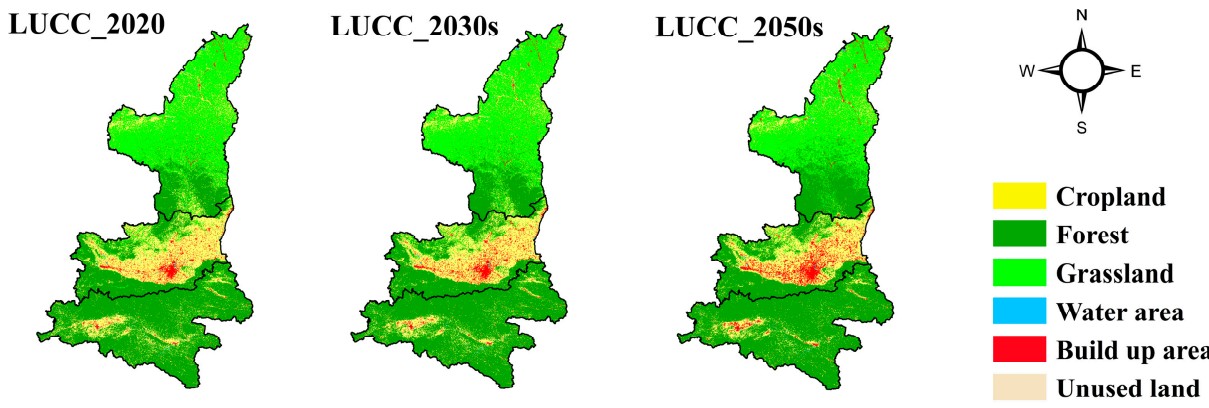

**Figure 11.** Spatial Distribution of Land Use in Shaanxi Province in 2020, 2030 and 2050.

It can be seen from Figure 11 that the cropland, forest, and built up areas in northern Shaanxi would increase year by year, and the unused land would decrease; the area of cropland and grassland in Guanzhong would decrease, while the forest and built up areas would increase; the area of cropland and grassland in southern Shaanxi would decrease, and the forest and built up areas would increase. Generally, the proportion of cropland, grassland, and unused land in Shaanxi Province would decrease, and the proportion of forest, built up and water areas would increase. The reduction in cropland and grassland areas indicate that Shaanxi Province would have achieved remarkable progress and effective results in the project of returning cropland to forests; the increase in built up areas and the reduction in unused land means that the urbanization process of Shaanxi Province would continue to advance and the economy would continue to be of high quality in the future. The increase in forest and water areas also reflects the strategic measures and layout of Shaanxi Province for the sustainable development of the ecological environment.

### 3.5. Impact of Future Precipitation and Land Use Change on Water Yield Services

3.5.1. Influence of Precipitation Change on Water Yield

The change in precipitation has a direct impact on water yield. Under the scenario of precipitation change, land use will not change. Based on the 2020 land use, this paper discussed the changes in water yield in Shaanxi Province in the 2030s and 2050s under the carbon emissions scenarios of RCP2.6, RCP4.5, and RCP8.5.

Compared with the real scenario in 2020, the annual water yield of Shaanxi Province in the 2030s under the carbon emission scenarios of RCP2.6, RCP4.5, and RCP8.5 is predicted to be 257.39 mm, 253.37 mm and 271.36 mm, with an increase of 59.58 mm, 55.57 mm, and 73.55 mm, and the annual average growth rate will be 3.01%, 2.81% and 3.72%, respectively. In the 2050s, under the carbon emission scenarios of RCP 2.6, RCP 4.5, and RCP 8.5, the annual water production of Shaanxi Province is predicted to be 273.37 mm, 269.28 mm and 304.41 mm, with an increase of 75.57 mm, 71.48 mm and 106.6 mm, and the average annual growth rate will be 1.27%, 1.2% and 1.79%, respectively. The water yield under

different emission scenarios is in the order of RCP 8.5 > RCP2.6 > RCP4.5. Compared with the carbon emission scenarios of RCP 2.6 and RCP 8.5, the probability of extreme weather events, including extreme rainfall events, is low under the carbon emission scenario of RCP 4.5; therefore, the change range of water yield is relatively small, which is consistent with the change characteristics of precipitation under different carbon emission scenarios in the future, indicating the synergistic relationship between water yield and precipitation, The water yield increases with the increase of precipitation. From 2020 to the 2030s, and then to the 2050s, the annual average change rate of water yield is predicted to decrease, and the change in water yield is predicted to be in a growing state with a tendency to be stable.

The spatial distribution of water yield under the carbon emission scenarios of RCP2.6, RCP4.5, and RCP8.5 in Shaanxi Province is shown in Figure 12. The spatial distribution of water production in Shaanxi Province in the 2030s and 2050s still shows the characteristics of "more in the south and less in the north". The water production in southern Shaanxi still ranks highest in Shaanxi Province, followed by Guanzhong and northern Shaanxi. The water yield under the carbon emission scenarios of RCP 2.6, RCP 4.5, and RCP 8.5 in Northern Shaanxi in the 2030s will be 43.74 mm, 44.75 mm, and 51.44 mm; 134.04 mm, 134.34 mm and 146.17 mm in Guanzhong; and 597.27 mm, 584.14 mm, and 619.81 mm in Southern Shaanxi. The change characteristics between different regions in Shaanxi Province in the 2050s will be consistent with that in the 2030s. The high value area is located in the Jialing River basin and the Han River basin in southern Shaanxi; the median-value area is located in the Jinghe River and Weihe River basins in the Guanzhong Plain, the Luohe River basin and the Yanhe River basin in northern Shaanxi and the Shaanxi border of the main stream of the Yellow River; and the low-value area is located in high-altitude areas such as Baiyu Mountain in northern Shaanxi.

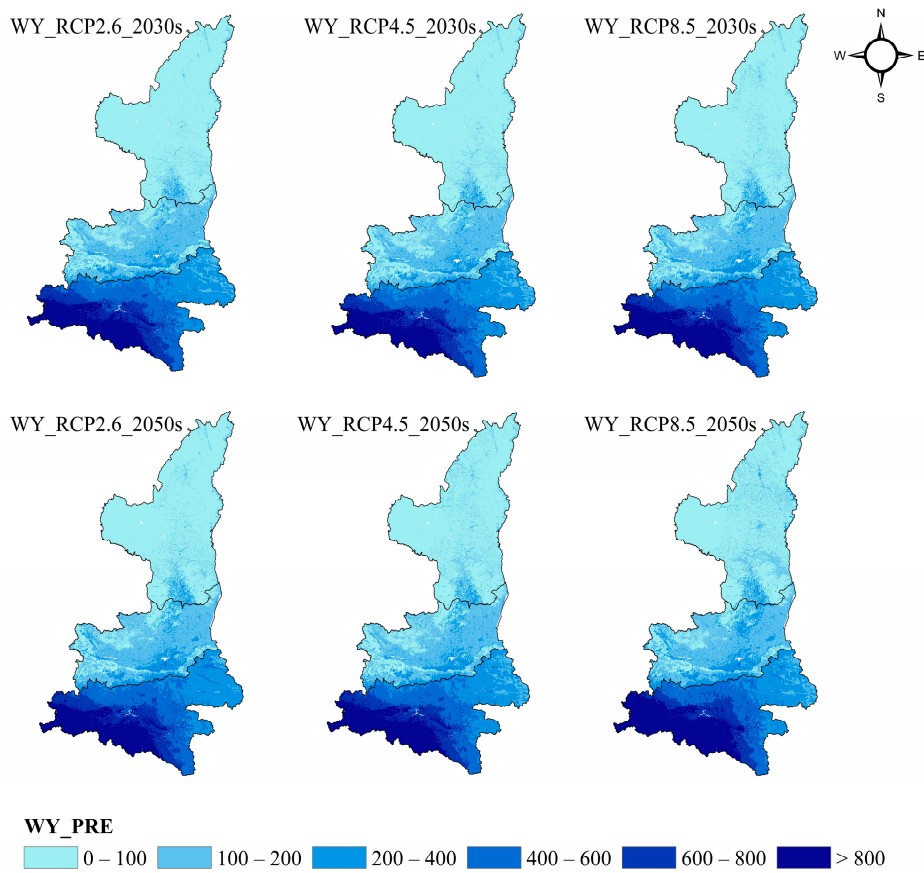

**Figure 12.** Spatial distribution of water yield under the situation of precipitation change in Shaanxi Province.

### 3.5.2. Influence of Land Use Change on Water Yield

Land use will change the regional water cycle, and affect the vegetation evapotranspiration, water infiltration process and soil water holding capacity, so as to change the water yield. Under the land use change scenario, the precipitation did not change. Based on the multi-year precipitation from 2000 to 2020, the simulated land use data of Shaanxi Province in 2030 and 2050 were used to discuss the changes in the water yield of Shaanxi Province in 2030 and 2050.

From 2000 to 2020, the average annual water production depth of Shaanxi Province was 191.95 mm, including 41.39 mm in northern Shaanxi, 133.16 mm in Guanzhong and 401.31 mm in southern Shaanxi. Compared with 2000–2020, the water production depth of Shaanxi Province in 2030 and 2050 will decrease by 11.9 mm and 14.14 mm, including 8.13 mm and 11.21 mm in northern Shaanxi, 3.86 mm and 0.62 mm in central Shaanxi, and 14.4 mm and 20 mm in southern Shaanxi, respectively. Under the change in land use, Shaanxi Province as a whole and in terms of the three regions will show downward changes in 2030 and 2050. In addition to the increase in the Guanzhong region in 2050 compared with 2030, both northern and southern Shaanxi regions will show a continuous downward trend.

According to the spatial distribution map of water yield in Shaanxi Province under the land use change scenario (Figure 13), compared with the precipitation change scenario, the high-value area of water yield in Shaanxi Province will transfer from the Jialing River basin in the southwest of Shaanxi Province to the Daba Mountain area in Ankang City, and the overall change in water yield will be small. The proportion of areas with a water yield depth of >800 mm will be relatively small, and the changes in the median and low value areas is predicted to be small.

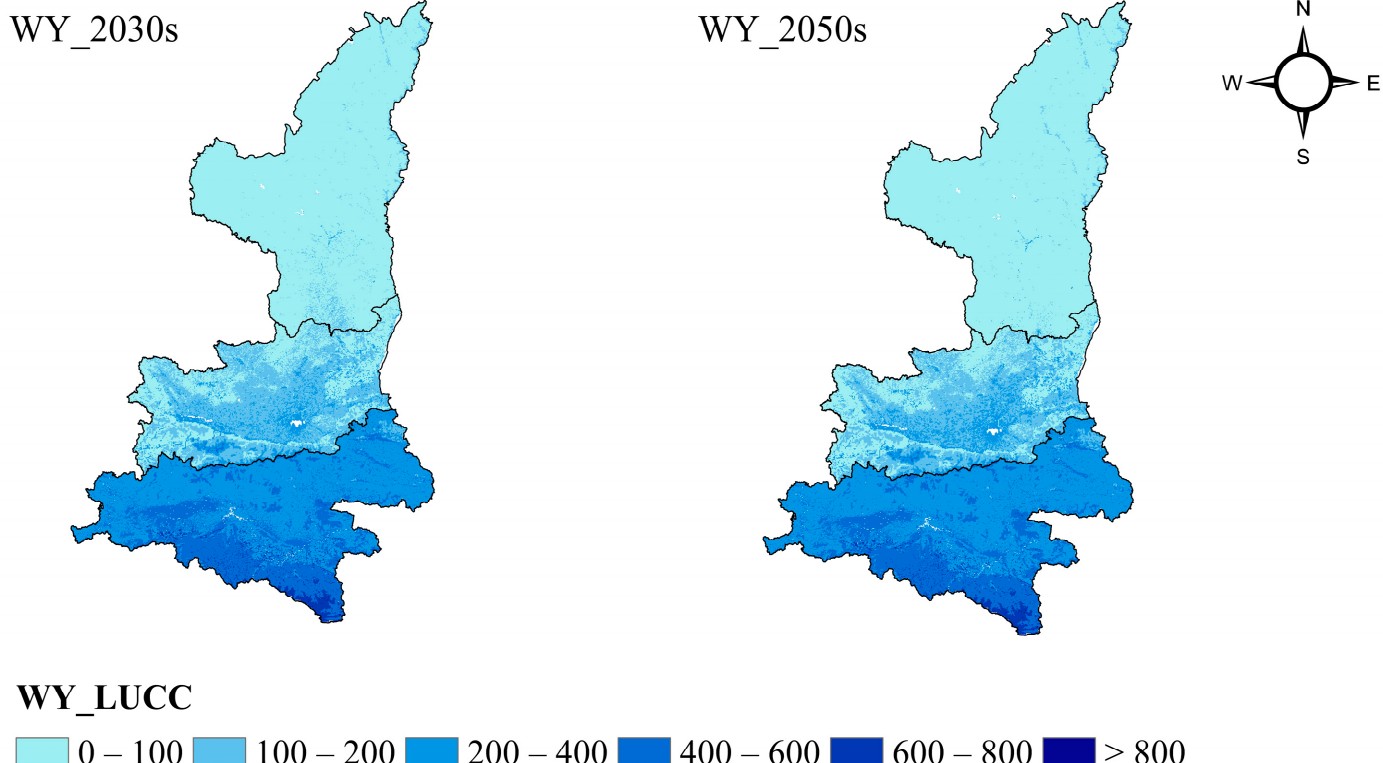

**Figure 13.** Spatial distribution of water yield under the situation of land use change in Shaanxi Province.

### 3.5.3. The Contribution Rate of Precipitation Change and Land Use Change Scenarios

The contribution rate of precipitation change and land use change scenarios to water yield is shown in Table 5.

**Table 5.** Contribution rate of precipitation change and land use change scenarios to water yield.

| Scenarios | Area | 2030s | 2050s |
|---|---|---|---|
| | Northern Shaanxi | 0.29 | 0.63 |
| Precipitation change scenario | Guanzhong | 0.23 | 0.94 |
| | Southern Shaanxi | 0.93 | 0.91 |
| | Northern Shaanxi | 0.71 | 0.37 |
| Land use change scenario | Guanzhong | 0.77 | 0.06 |
| | Southern Shaanxi | 0.07 | 0.09 |

Under the precipitation change scenario, the change in water yield depth in Shaanxi Province increased with time, and the change in precipitation had a positive growth effect on the change in water yield depth. The contribution rate of precipitation change to southern Shaanxi in the 2030s and 2050s will be more than 90%, which will play a decisive role relative to land use. In the 2030s, the contribution rate of precipitation change in Northern Shaanxi and Guanzhong will be approximately 30%; in the 2050s, the contribution rate of precipitation change to water yield change in Northern Shaanxi and Guanzhong will be approximately about 70%. It can be seen that, under the situation of precipitation change, the contribution rate of precipitation change to water yield change in northern Shaanxi and Guanzhong gradually increased. For southern Shaanxi, precipitation has always played a leading role in terms of land use.

Under the land use change scenario, the depth of water production in Shaanxi Province decreased with time change. Compared with 2020, the depth of water production in northern Shaanxi, Guanzhong and southern Shaanxi is predicted to be lower. From the perspective of contribution rate, in the 2030s, the contribution rate of land use change to water yield change in northern Shaanxi and Guanzhong will exceed 70%; in the 2050s, the contribution rate of land use change to water yield change in northern Shaanxi and Guanzhong will decrease to 30%. The contribution rate of land use change to water yield change in Southern Shaanxi in the 2030s and 2050s will be less than 10%.

*3.6. Discussion*

3.6.1. Correlation Analysis between Different Calculation Parameters and Water Yield

Nine influencing factors were selected for correlation analysis with water yield in Shaanxi Province (Figure 14). As a result of the correlation coefficient between precipitation and water yield being 0.897, and the variation characteristics of precipitation and water yield in Shaanxi Province from 2000 to 2020, it can be confirmed that precipitation was the most vital factor for water yield services and important in terms of precipitation to model the input, which was consistent with the findings of Cao et al. [57]. From the correlation analysis results, the actual evapotranspiration and potential evapotranspiration were negatively correlated with water yield. According to the water balance principle, the difference between actual evapotranspiration and precipitation was the water retention of the ecosystem, while potential evapotranspiration and actual evapotranspiration were closely related; therefore, both actual evapotranspiration and potential evapotranspiration also had an impact on water yield. LAI could characterize the vegetation coverage on the land surface and was an essential structural parameter of the ecosystem [58], which had an important influence on the exchange of matter and energy between vegetation, land surface, and the atmosphere [59]. The correlation coefficient between LAI and water yield was 0.658 ($p < 0.01$). In general, LAI is not only proportional to the density of vegetation, but also to the vegetation coverage area. In other words, LAI increases with the growth of vegetation coverage. With the increase of vegetation coverage, the water conservation capacity of the forest will be higher. When the forest land coverage is large enough, the hydrological conditions of the soil will be improved. The amount of water intercepted by the leaves and the amount of water consumed by evapotranspiration will be much smaller than the amount of water left in the forest ecosystem. Accordingly, LAI can show a positive correlation with the water yield. In addition, we also found that the correlation coefficient

between the depth of vegetation roots and water yield reached 0.465 ($p < 0.01$), but root depth was negatively correlated with water yield. The deeper the root of the vegetation, the stronger its water absorption capacity, and part of the water would be absorbed into the plant, thereby reducing the regional water yield. The correlation coefficient between water yield and soil type in Shaanxi Province was 0.454 ($p < 0.01$). The results showed that there was a strong correlation between water yield and soil types. Different soil types have certain effects on the spatial distribution of water yield. In each land use type in Shaanxi Province (Figure 3), the water yield depth of ferruginous soil, eluvial soil and alpine soil is 445.34 mm, 365.39 mm and 352.86 mm, respectively, which is the soil type with high water yield distribution; the water yield depth of saline-alkali soil, hydrous soil and arid soil is 55.93 mm, 25.75 mm and 13.47 mm, respectively, which is the soil type with low water yield distribution. Finally, we also considered the influence of terrain factors, temperature, and other factors on the water yield of Shaanxi Province. Compared with evapotranspiration and precipitation, they had less of an impact on the water yield of Shaanxi Province.

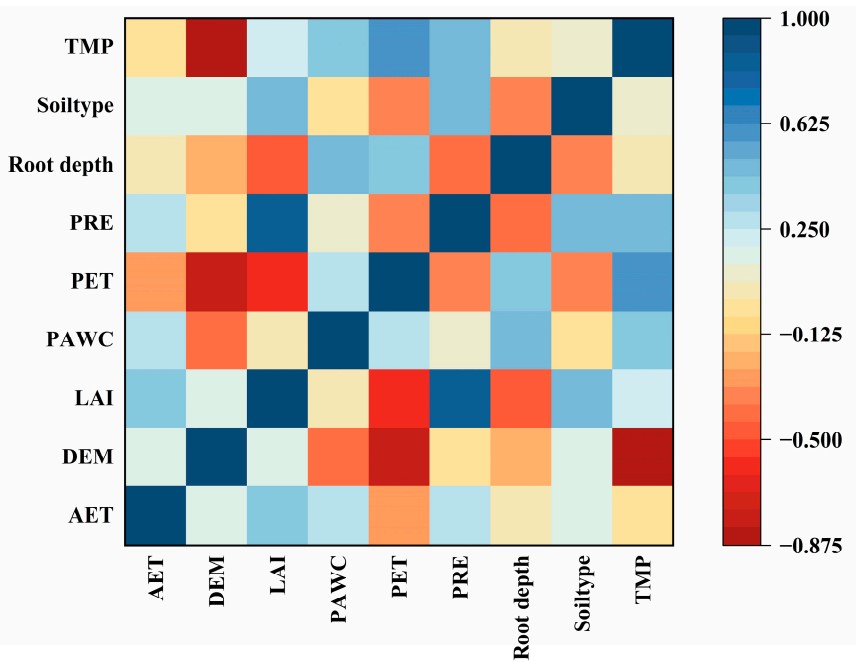

**Figure 14.** Correlation coefficients between influencing factors and water yield. Notes: All correlations are significant at the 0.01 level.

3.6.2. Calibration and Limitations of Water Yield Assessment

The Budyko dryness index theory showed that the higher the Z value, the less the model results were affected by the seasonal constant Z [25]. In this paper, we mainly used the total water resources data from the Shaanxi Provincial Water Resources Bulletin to calibrate the model parameters. According to the Shaanxi Provincial Water Resources Bulletin, the annual average total water resources in northern Shaanxi, Guanzhong and southern Shaanxi was $33.24 \times 10^8$ m$^3$, $73.75 \times 10^8$ m$^3$, and $280.63 \times 10^8$ m$^3$, respectively. The water yield of the three regions in Shaanxi Province was obtained by changing the size of the Z value and repeatedly running it. The results were compared with the recorded total water resources, and a Z value was selected. Finally, the difference between the water yield obtained using this value and the total water resources was the smallest. After many adjustments and repeated verifications, when the Z values were 15.7, 14.29, and 3.55, the water yield was highly consistent with the actual total water resources, the overall error was the smallest, and the model simulation effect was the best. Figure 15 shows the comparison results of the recorded total water resources and simulated water yield in northern Shaanxi, Guanzhong, and southern Shaanxi. We used the decisive coefficient R$^2$ to judge the correlation between simulation and actual measurement. The decisive coefficients

$R^2$ of northern Shaanxi, Guanzhong and southern Shaanxi were 0.625, 0.705, and 0.689, respectively. The results showed that the simulation effect was good and has credibility.

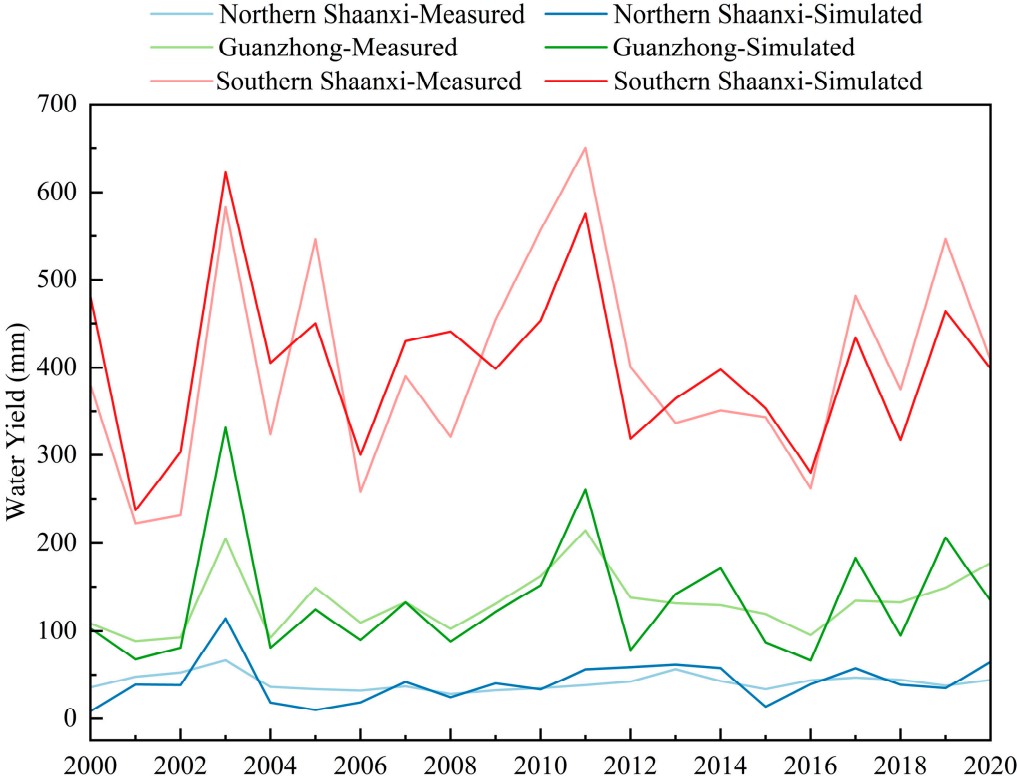

**Figure 15.** The comparison results of recorded total water resources and simulated water yield in northern Shaanxi, Guanzhong, and southern Shaanxi.

In addition to the Z value, although this study provides convenience for assessing water yield services in Shaanxi Province, there are still limitations and uncertainties. First, the data input into the InVEST model, including precipitation and potential evapotranspiration, were annual average data. This does not take into account extreme conditions and seasonal changes in water yield. Moreover, the data and parameters input by the water yield module also included data such as the depth to root-restricting layer, plant available water fraction, and the plant evapotranspiration coefficient, which came from the Chinese soil dataset, parameters recommended by the Food and Agriculture Organization of the United Nations (FAO), and multiple experiments to verify the data. There was a large degree of uncertainty in the unified standard and applicability evaluation, and the simulation accuracy of the model needs to be confirmed by combining more actual observation data. In addition, future data simulations were based on historical data and forecasting methods, but they have implications for future forecasting [60]. When forecasting the future water yield, only future precipitation and land use were simulated, and only RCP scenarios caused by greenhouse gas concentrations were considered. Finally, the research was conducted on a provincial scale. Due to the great differences in the natural environment and social and economic conditions in the province, the water yield simulations were carried out from northern Shaanxi, Guanzhong, and southern Shaanxi. Only macro simulations and analyses were carried out, and there was a lack of microscopic exploration at a smaller scale. In subsequent research, if necessary, we could improve the accuracy of the data, narrow the scope of the study, make the water yield status of small-size and small watersheds clearer, and provide scientific support for the arrangement of activities related to production and living.

### 3.6.3. Reflections on the InVEST Model

As far as the InVEST model is concerned, it is currently widely used in large-scale areas, and data with low precision are used to evaluate the large-scale ecosystem services. Previous studies have considered the spatial distribution characteristics of ecosystem services and the value of various ecosystem services under different land use types; however, due to the low precision of large-scale data, it is impossible to accurately assess the distribution characteristics of ecosystem services under different vegetation types. Therefore, it is hard to accomplish optimized and enhanced ecosystem services. In addition, the InVEST model is one of the main models for the evaluation of ecosystem services. It includes three structures of freshwater ecosystems, marine ecosystems, and terrestrial ecosystems, with carbon storage, water yield, habitat quality, nutrient deposition, aquaculture, soil conservation, timber production and other modules; however, those currently used in China are limited to modules such as carbon storage, water production, habitat quality, soil conservation, etc. The utilization rate of the InVEST model modules is low, and it is still necessary to continue to expand the research category of and improve the research on ecosystem services in the future.

Since the InVEST model does not distinguish surface water, groundwater, and base flow, the change in precipitation can be directly reflected by the change in water yield. Therefore, it is impossible to assess the specific impact of runoff and groundwater on the change in water yield. On the one hand, the increase in precipitation will increase the risk of regional water and soil loss, and on the other hand, it will increase the regional water supply. The water yield module of the InVEST model only considers the impact of precipitation on the increase in water yield. In the future, the impact of precipitation intensity and precipitation frequency on regional water source occurrence should be included in the modification and improvement of the model.

### 4. Conclusions

Based on soil data, meteorological data, remote sensing data, and socioeconomic data, this study estimated and analyzed the water yield and its temporal and spatial variation characteristics in Shaanxi Province from 2000 to 2020 by using the InVEST model water yield module. The precipitation and land use in Shaanxi Province in the 2030s and 2050s were predicted, and the change characteristics of water yield under the scenarios of precipitation and land use change were analyzed. The contribution rate of precipitation and land use change to the change in water yield in Shaanxi Province was quantitatively analyzed using the scenario analysis method. The following conclusions were drawn: from 2000 to 2020, the multi-year average water yield in Shaanxi Province was $128.3 \times 10^8$ m$^3$. The water yield pattern in Shaanxi Province showed obvious temporal and spatial heterogeneity. Under the precipitation change scenario, the water yield under different emission scenarios was in the order of RCP8.5 > RCP2.6 > RCP4.5; under the land use change scenario, the water yield depth of Shaanxi Province as a whole and in terms of the three regions showed a predicted decline in the 2030s and 2050s.

The research on the service of the water ecosystem in Shaanxi Province has great significance for the allocation of water resources, optimization of water ecosystem structure, and sustainable utilization of water resources. By dividing Shaanxi Province into three regions and calculating the corresponding water yield, it is possible to monitor the difference in water yield in different geographical environments in Shaanxi Province, and to understand the influence of the geographical environment on water yield, which can also provide a reference for the study of water ecosystem services in larger geographic areas. Monitoring the contribution of precipitation and land use changes to water yield changes can quantitatively study the impact of precipitation and land use changes on water yield.

This paper used 1 km resolution data to evaluate the water yield service of Shaanxi Province. In the future, it is still necessary to improve the data accuracy and evaluate the ecosystem services of Shaanxi Province on a small scale. Furthermore, relevant studies of influencing factors can be added.

**Author Contributions:** Y.H., conceptualization, writing—review and editing; Y.L., formal analysis, writing—original draft; W.L. and L.J., data curation; Y.Z., methodology. All authors have read and agreed to the published version of the manuscript.

**Funding:** This research was jointly supported by the National Science and Technology Basic Resource Investigation Program (Grant No. 2017FY100904), the China Postdoctoral Science Foundation (Grant No. 2018M633602), the Postdoctoral Research Fund of Shaanxi Province (Grant No. 2017BSHEDZZ144), and the Natural Science Basic Research Plan in Shaanxi Province of China (Grant No. 2021JQ-449).

**Data Availability Statement:** Not applicable.

**Acknowledgments:** We acknowledge the data support from the "National Earth System Science Data Center, National Science and Technology Infrastructure of China. (http://www.geodata.cn, accessed on 3 May 2022)" and other websites and platforms that provide data. We would like to thank the anonymous reviewers and editors for their valuable comments and suggestions.

**Conflicts of Interest:** The authors declare that they have no known competing financial interests or personal relationships that could have appeared to influence the work reported in this paper.

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
