# Peer review of "Evaluation and Prediction of Water Yield Services in Shaanxi Province, China"

_forests, doi:10.3390/f14020229_

Round 1

Author Response

Dear Reviewer,

Thank you for your helpful comments on our manuscript. We read and discussed these comments carefully and revise our manuscript accordingly. The main corrections in the paper and the response to the comments are as following.

Major Comments:

Point 1I have missed a thorough revision about the physical aspects of water yield in the Introduction. How is water yield improved in some ecosystems? What are the physical processes underneath it? How infiltration and/or runoff generation can be modified by different land uses? Is it interesting to improve infiltration or runoff? You must provide an overview of these conditions to highlight the benefits of ecosystem conservation.

Response 1: According to reviewer’ comments, we added the following in the introduction section: How is water yield improved in some ecosystems? What are the physical processes underneath it? How infiltration and/or runoff generation can be modified by different land uses? and the benefits of ecosystem conservation.

Physical process of water yield:

The InVEST model water yield module assumes that the water yield of each grid cell is collected at the watershed outlet in the form of runoff, and the water supply is the water yield after subtracting the actual evapotranspiration from the precipitation of each grid cell. The model does not distinguish surface water, groundwater and base flow. We had added the manuscript in the line 55-58.

How is water yield improved in some ecosystems?

Through the use of data in the water yield module of the InVEST model and the analysis of existing studies, water yield and climate factors are closely related to land use. Therefore, we can improve regional water yield by indirectly changing the way of land use and intervening in climate factors. For example, we can increase regional water yield by increasing vegetation coverage, reducing bare land area, reducing evapotranspiration, and improving water conservation capacity. We had added the manuscript in the line 84-90.

The impacts of different land use types on runoff or infiltration are as follows:

The built up land is mainly composed of hardened pavement and buildings. Most of the surface water formed by precipitation will converge into runoff, while the cultivated land, grassland and forest land are mainly composed of vegetation. Compared with built up land, it has a certain water interception capacity, and the runoff formed by precipitation will be correspondingly less.

Through the concept of "sponge city" proposed previously, the construction land can increase the water conservation capacity of urban construction area and change the surface water infiltration. Among the cultivated land, forest land and grassland, the water conservation capacity of forest land is the strongest, and the water infiltration can be changed through the transformation of cultivated land and grassland to forest land.

This content had been added the manuscript in the line 91-96 and 99-103.

The benefits of ecosystem conservation:

Through the transformation between different land uses, the water yield can be adjusted, which is beneficial to the stability of water resources and the balance of ecosystems. We had added the manuscript in the line 96-98.

Point 2The downscaling model was calibrated using observed meteorological data. However, the information regarding the meteorological station was not provided in the M&M section. Where all the three regions properly covered? How many stations do you have in each one?

Response 2: According to reviewer’ comments, the relevant information of 46 meteorological stations in Shaanxi Province used in the downscale model was added in the M&M section. Due to the different completeness of daily precipitation data of different meteorological stations, we selected 46 stations with complete daily precipitation data from 1961 to 2005 from all meteorological stations in Shaanxi Province, including 11 stations in northern Shaanxi, 20 stations in Guanzhong and 15 stations in southern Shaanxi. Due to the distribution of the central plain in Guanzhong area, there are many meteorological stations with high data integrity, so there are relatively many stations available for selection. The specific location of the site can be found in Figure 1. We had added the manuscript in the line 238-244.

Point 3Neither presented the land use map for 2020 (actual scenario) nor the spatial distribution of rainfall in the 2000-2020 period. These maps are important for understanding the effects of the spatial distribution of land use and precipitation on the water yield presented in Figure 4. The map of actual evapotranspiration is also important (200-2020). With all this information, the reader can have an overall understanding of the water yield (driving forces) in the observed period (and then be able to observe patterns in the climate change and land use changes scenarios).

Response 3: According to reviewer’ comments, in the result section, "3.2 Spatial distribution of precision and actual evapotranspiration from 2000 to 2020" was added, followed by describing the spatial change characteristics of precipitation and actual evapotranspiration in Shaanxi Province, and the spatial distribution of land use in Shaanxi Province in 2020 was added in Figure 11.

Point 4Table 3 must be in the M&M section describing the predictors used in the SDSM model together with the information about their source. It should be described in the M&M section to improve understanding of the study development.

Response 4: According to reviewer’ comments, we added the source and selection process of predictors in "2.2.2 SDSM Model" in the M&M section. The specific content added in the paper is "The predictors are derived from the predictors of historical and future climate change scenarios in the large-scale grid corresponding to Shaanxi meteorological stations in the CanEsm2 dataset developed by the Canadian Environmental and Climate Change Modeling and Analysis Center in CMIP5. Through the analysis of the explained variance and partial correlation coefficient between the daily precipitation data of Shaanxi meteorological stations and 26 predictors, 3 - 5 predictors with the highest correlation with precipitation are obtained, which can be used to predict precipitation in future climate change scenarios".

Point 5How much of the increased water yield will be due to runoff? And to groundwater flow? Increased precipitation (with likely more intense events in the climate change scenarios) might increase soil erosion and runoff (i.e., soil loss and floods). Although more precipitation will produce more water, it does not mean a well-preserved watershed. You should provide a critical analysis in this regard.

Response 5: Since the InVEST model does not distinguish surface water, groundwater and base flow, the change of precipitation can be directly reflected in the change of water yield. Therefore, it is impossible to assess the specific impact of runoff and groundwater on the change of water yield. The increase of precipitation will increase the risk of regional water and soil loss on the one hand, and increase the regional water supply on the other hand. The water yield module of InVEST model only considers the impact of precipitation on the increase of water yield. In the future, the impact of precipitation intensity and precipitation frequency on regional water source occurrence should be included in the modification and improvement of the model. We had added the manuscript in the line 715-723.

This study only analyzed the contribution rate of precipitation and land use change to the change of water yield. In this part, we should add the specific change of water yield caused by precipitation change. This part was described in "3.4.1 Influence of precision change on water yield ".

Point 6 Missing a discussion regarding soil’s importance to water yield. You used soil depth as an input variable in InVEST but did not provide any information regarding the soils in the Shaanxi region. Is there a difference in soil’s characteristics that caused the spatial distribution observed in water yield?

Response 6: According to reviewer’ comments, we added the relevant information of Shaanxi soil and the soil type distribution (Figure 2). In the discussion part, we obtained that the correlation coefficient between water yield and soil type in Shaanxi Province was 0.454. The results showed that there was a strong correlation between water yield and soil types. Different soil types have certain effects on the spatial distribution of water yield. We had added the manuscript in the line 643-646.

Minor comments:

Point 1Line 43-44: This information has already been provided. You can remove it.

Response 1: According to reviewer’ comments, repeated content has been deleted. "It plays a key role in improving regional hydrological conditions and regulating regional water cycle".

Point 2Line 66-68: What did these studies highlight? It will be interesting if you describe the main findings of these studies.

Response 2: According to reviewer’ comments, the following content was added: "Through the InVEST model water yield module, this study mainly obtained the temporal and spatial changes in water yield in Shaanxi Province from 2000 to 2020 and the impact of future precipitation and land use changes on water yield changes". We had added the manuscript in the line 72-74.

Point 3Line 177: You should provide the meaning of GCM before using the acronym.

Response 3: The full name of GCM is "Global Climate Model", which is widely used to predict the future climate and is the main tool to predict the future evolution of the atmosphere at different time scales. We added the full name of GCM in the line 220.

Point 4Line 188-189: "The IPCC recommends that the base period could be longer than 30 years". This sentence sounds weird. Maybe rewrite it as "The IPCC recommends base periods longer than 30 years, so …".

Response 4: According to reviewer’ comments, "The IPCC recommends that the base period could be longer than 30 years" was rewritten as "The IPCC recommends base periods longer than 30 years, so this study selects 30 years to describe the climate characteristics of the region".

Point 5 Line 270: The average annual precipitation or the annual precipitation?

Response 5: According to reviewer’ comments, "the average annual precipitation" was rewritten as "the annual precipitation".

Point 6Line 271-275: In the sentence, it seems that the increase in precipitation only improves evapotranspiration, i.e., increased precipitation did not affect the water yield. But you also had an improved water yield (see years 2013, 2011, and 2017 in Figure 2). Please, correct this.

Response 6: The precipitation and water yield in Figure 2 used the "Y axis" of different starting points, so the change of precipitation was opposite to that of water yield. According to the change curve of precipitation and water yield from 2000 to 2020, it can be seen that the change characteristics of water yield and precipitation are consistent, including 2011-2013 and 2017. The precipitation in 2013 and 2017 was higher than that in adjacent years, and the water yield shows the same rule. Now Figure 2 has been changed to Figure 4. For easier understanding, Figure 4,the axe of precipitation, potential, and actual evapotranspiration was presented in a same orientation.

Point 7 Line 342-342: Where are the stations with poor classification (spatially)? And the good ones? Was there a worst behavior in a specific region?

Response 7: According to reviewer’ comments, we added location information of sites with good simulation effect and sites with poor simulation effect. Zichang Station and Pingli Station have the worst simulation effect, with their NS lower than 0.5, respectively located in northern and southern Shaanxi. Shenmu, Zhidan, Lintong, Luochuan and Xi'an Stations have the best simulation effect, with their NS greater than 0.93. Shenmu, Zhidan and Luochuan Stations are located in northern Shaanxi, and Lintong and Xi'an Stations are located in Guanzhong. There was no obvious spatial correlation between good and bad sites. We had added the manuscript in the line 452-457.

Point 8Line 385: A brief description of the Markov Chain method is needed in the M&M section.

Response 8: According to reviewer’ comments, we added relevant contents about Markov chain in "2.2.3 PLUS Model" of M&M section.

Markov method: Markov method is an analysis method that uses the current situation and trend of a variable to predict the future state and trend. The mutual transformation of different land use types in the region is a complex process, which is difficult to accurately describe with functional relationship, that is, under certain conditions, the change of land use types conforms to the nature of Markov random process, so it is feasible to use it to study the dynamic transformation of land use types, and the mutual transformation between land types can be quantitatively obtained. We had added the manuscript in the line 276-282.

Point 9Line 444: Spatial or temporal variation of water yield?

Response 9: What the article wants to express is temporal variation of water yield, which has been corrected.

Point 10Line 475-478: I am not sure about the reasons for the positive correlation between LAI and water yield. Greater values of LAI are related to dense forests, which have greater amounts of intercepted water and evapotranspiration (i.e., more water returns into the atmosphere. See the negative correlation with root depth). However, vegetated areas increase water yield due to improved soil hydrology (i.e., preferential flows, infiltration, water retention, litterfall, …). Moreover, evaporation from the soil has little importance when compared to (or even within) vegetated areas. In summary, forests and grasslands improve soil characteristics, increasing water infiltration and percolation.

Response 10: The positive correlation between LAI and water yield can be explained as follows: in general, LAI is not only proportional to the density of vegetation, but also to the vegetation coverage area. In other words, LAI increases with the growth of vegetation coverage. With the increase of vegetation coverage, the water conservation capacity of the forest will be higher. When the forest land coverage is large enough, the hydrological conditions of the soil will be improved. The amount of water intercepted by the leaves and the amount of water consumed by evapotranspiration will be much smaller than the amount of water left in the forest ecosystem. Accordingly,LAI can show a positive correlation with the water yield.

Point 11Line 492-494. This information of Z must be in the M&M section.

Response 11: According to reviewer’ comments, we have put the information about Z value in M&M section "2.3.1 Evaluation method of the water yield services".

Point 12Maybe a flowchart describing the phases of model development and application.

Response 12: Flowchart was added to the M&M section, which was reflected in the article "2.6 Research ideas".

Point 13Line 503: What did you mean with "measured water resources"? How it was "measured"?

Response 13: "measured water resources" means total actual water resources recorded in Shaanxi Water Resources Bulletin。

Point 14Figure 3: It would be interesting having information on precipitation in each region. We would be able to understand the importance of the precipitation spatial distribution on the amount of water yield by analyzing precipitation X water yield at once.

Response 14: According to reviewer’ comments, we added three regional precipitation curves of Shaanxi Province in Figure 3. The water yield of three regions in Shaanxi Province is on the left, and the precipitation is on the right. Since it was difficult to distinguish water yield and precipitation when they were expressed together, they were expressed separately. Now Figure 3 in the article has been modified to Figure 5.

Point 15Equation 1: what "J" means? And "x". It must be defined.

Response 15: "J" and "i" are added to the formula. Where "j" refers to the jth land use type and "x" refers to a grid.

Point 16Equations 2 and 3 need formatting.

Response 16: According to reviewer’ comments, all formulas have been reedited to conform to the required format.

Reviewer 2 Report

The paper presents the use of models to understand the impacts of changes in precipitation regimes and land use on water yield. It presents an innovative perspective and combined methods that produce interesting results, such as the spatial and temporal hydrological behavior of water yield.

However, there are some observations:

Calculating the annual water yield rate can improve the presentation of results.

Are the precipitation data used in the survey measured by the same method? Make this clear to readers outside your home country.

Figure 4: Are the values presented relative to the sum of 5 years?

The validation of simulations and data methods should be better explored in the paper.

Author Response

Dear Reviewer,

Thank you for your helpful comments on our manuscript. We read and discussed these comments carefully and revise our manuscript accordingly. The main corrections in the paper and the response to the comments are as following.

Point 1Calculating the annual water yield rate can improve the presentation of results.

Response 1: In this study, the InVET module "Annual Water Yield" is used to calculate the water yield of Shaanxi Province from 2000 to 2020. The annual water yield of Shaanxi Province is obtained by running the model. In the result part, in order to better show the interannual change of water yield in Shaanxi Province, five years were used as a cycle for display.

Point 2Are the precipitation data used in the survey measured by the same method? Make this clear to readers outside your home country.

Response 2: The precipitation data used in the calculation of water yield from 2000 to 2020 in this study is from the monthly precipitation data set of China from 1901 to 2021 provided by China National Earth System Science Data Center. We added "3.2 Spatial distribution of precision and actual precipitation from 2000 to 2020" to the results of the precipitation data results of Shaanxi Province from 2000 to 2020 obtained through cutting and synthesis, and made a feature analysis. When downscaling the future precipitation data of Shaanxi Province, the daily precipitation data of the stations input by the SDSM model is from the Chinese surface climate data set V3.0.

Point 3Figure 4: Are the values presented relative to the sum of 5 years?

Response 3: Figure 4 the spatial distribution map of water yield in Shaanxi Province was made with a cycle of 5 years and the results were described with a cycle of 5 years. Figure 4 has been modified to Figure 7.

Point 4The validation of simulations and data methods should be better explored in the paper.

Response 4: According to reviewer’ comments, we will improve the validation of simulations and data methods, including more detailed explanation of methods and results validation.

Reviewer 3 Report

·       The authors should improve the formulation of the originality and scientific novelty of their study.It is very important to clearly formulate what is going to be new/novel in this paper.

·        The Introduction section needs to bring a clear context to the study, i.e. to indicate in more details what has been done in the past in the sphere of using the InVest model for analyzing the water yield and its future changes.

·       Instead of the “geographic models“ in the sentence: “With the continuous progress and upgrading of remote sensing and geographic information system (GIS) technology, geographic models have played an essential role in the fields of ecology and Hydrology” (Lines 51-53), it is better to write: “hydrological models“.“Hydrology” should be written with small caps.

·       There are many important papers in which the joint application of hydrological models, GIS and remote sensing data were presented and greatly contributed to the better understanding of hydrological and ecological processes in catchments. Therefore, the sentence: “With the continuous progress and upgrading of remote sensing and geographic information system (GIS) technology, geographic models have played an essential role in the fields of ecology and Hydrology” (Lines 51-53)  should be supported by citing of additional papers and researches of this topic, such as:

 1.      Đukić, Vesna, Erić, Ranka, Dumbrovsky, Miroslav and Sobotkova, Veronika. "Spatio-temporal analysis of remotely sensed and hydrological model soil moisture in the small Jičinka River catchment in Czech Republic" Journal of Hydrology and Hydromechanics, vol.69, no.1, 2021, pp.1-12. https://doi.org/10.2478/johh-2020-0038

2.      Đukić, Vesna, and Ranka Erić. 2021. "SHETRAN and HEC HMS Model Evaluation for Runoff and Soil Moisture Simulation in the Jičinka River Catchment (Czech Republic)" Water 13, no. 6: 872. https://doi.org/10.3390/w13060872

·     The ranges of the average annual precipitation and evaporation in the study region should be inserted in the “2. Materials and Methods“ section. Also, the ranges of the altitudes in the study region should be written in this section (Lines 109-111)

·     For each parameter specified in the equations (1) – (7), the units in which the parameters are expressed should be written (Lines 133-165), also for the equations (8) and (9) (Lines 226-233). The labels of the equation numbers should be set appropriately.

·     It should be left a space between the numbers and its units

·     It should be left one empty row between the parahraphs and the equations, also between the paragraphs and the captions of the next sections, between the text and the table, between the figure and its caption

·     The caption of the Figure 8 is missing.

·     It would be better to incorporate the Section “3. Results“ and the Section “4. Discussion“ in one section: “3. Results and Discussion”.

·     In the “3. Results and Discussion” section it should be useful to link results obtained in this study with the results of some previous studies.

·     The subsection “4.3. Reflections on the InVEST Model” should be partly included in the ”4. Conclusion and partly in the 3. Results and Discussion section

·     Some corrections are marked in the text and the file with corrections is added as an attachment

·     In the end, the English language in the text is not good and it refers not only to the parts marked in the text but to the entire text. Therefore, it is necessary that the entire text be checked and corrected by a native English speaker or someone with good proficiency in English

Author Response

Dear Reviewer,

Thank you for your helpful comments on our manuscript. We read and discussed these comments carefully and revise our manuscript accordingly. The main corrections in the paper and the response to the comments are as following.

Point 1There are many important papers in which the joint application of hydrological models, GIS and remote sensing data were presented and greatly contributed to the better understanding of hydrological and ecological processes in catchments. Therefore, the sentence: “With the continuous progress and upgrading of remote sensing and geographic information system (GIS) technology, geographic models have played an essential role in the fields of ecology and Hydrology” (Lines 51-53) should be supported by citing of additional papers and researches of this topic.

Response 1: According to reviewer’ comments, we have quoted the following two articles in the paper.

  1. Đukić, Vesna, Erić, Ranka, Dumbrovsky, Miroslav and Sobotkova, Veronika. Spatio-temporal analysis of remotely sensed and hydrological model soil moisture in the small Jičinka River catchment in Czech Republic" Journal of Hydrology and Hydromechanics, vol.69, no.1, 2021, pp.1-12. https://doi.org/10.2478/johh-2020-0038
  2. Đukić, Vesna, and Ranka Erić. 2021. "SHETRAN and HEC HMS Model Evaluation for Runoff and Soil Moisture Simulation in the Jičinka River Catchment (Czech Republic)" Water 13, no. 6: 872. https://doi.org/10.3390/w13060872

Point 2The ranges of the average annual precipitation and evaporation in the study region should be inserted in the "2. Materials and Methods" section. Also, the ranges of the altitudes in the study region should be written in this section (Lines 109-111).

Response 2: The temporal and spatial characteristics of precipitation and evaporation in Shaanxi Province are added in the "3.2 Spatial distribution of precipitation and actual evapotranspiration from 2000 to 2020" section. The relevant content of altitude in Shaanxi Province is added in lines 135-138 of the paper.

Point 3For each parameter specified in the equations (1) - (7), the units in which the parameters are expressed should be written (Lines 133-165), also for the equations (8) and (9) (Lines 226-233). The labels of the equation numbers should be set appropriately.

Response 3: According to reviewer’ comments, we have changed the units of the equation parameters, the format of the equations, and the label of the equations.

Point 4RCP2.6、RCP4.5, and RCP8.5. It was not explained in the text what this abbreviation present.

Response 4: According to reviewer’ comments, we added the interpretation of climate change scenarios RCP2.6, RCP4.5, and RCP8.5 into the SDSM model. RCP is the abbreviation of "Representative Concentration Pathways", which measures the concentration of greenhouse gases in the atmosphere. RCP2.6 represents the path with the lowest greenhouse gas emissions, followed by RCP4.5, and RCP8.5 represents the path with the highest greenhouse gas emissions. We had added the manuscript in the line 218-223.

Point 5Maybe, it is better to write hydrological models instead of geographic models.

Response 5: According to reviewer’ comments, we changed the geographical models to the hydrological models. It is in line 50 of the text.

Point 6It should be written the equation that relates Kc and the Leaf Area Index (LAI).

Response 6: According to reviewer’ comments, we added Kc and LAI related formulas in "2.3.1. Evaluation method of the water yield services", as shown in Formula (4).

Point 7GCM: it was not explained what this abbreviation is.

Response 7: The full name of GCM is "Global Climate Model", which is widely used to predict the future climate and is the main tool to predict the future evolution of the atmosphere at different time scales. At the same time, we added the full name of GCM in the line 217.

Point 8Formula format and numbering.

Response 8: According to reviewer’ comments, all formulas have been reedited to conform to the required format.

Point 9It should be left a space between the numbers and its units.

Response 9: According to reviewer’ comments, we left a space between all the numbers and its units.

Point 10It should be left one empty row between the paragraphs and the equations, also between the paragraphs and the captions of the next sections, between the text and the table, between the figure and its caption.

Response 10: According to reviewer’ comments, we left one empty row between all the paragraphs and the equations, also between the paragraphs and the captions of the next sections, between the text and the table, between the figure and its caption.

Point 11The caption of the Figure 8 is missing

Response 11: The capation of Figure 8 has been added as "Spatial Distribution of Land Use in Shaanxi Province in 2020, 2030 and 2050", and now Figure 8 in the article has been modified as Figure 11.

Point 12"After many adjustments and repeated verifications, when the Z value was 15.7, 14.29, and 3.55, the water yield was highly consistent with the actual total water resources, the overall error was the smallest, and the model simulation effect was the best". Is it possible to explain this statement more precisely?

Response 12: According to reviewer’ comments, the adjustment of Z value was added as follows: the water yield of the three major regions in Shaanxi Province was obtained by changing the size of Z value and repeatedly running. The operation results were compared with the recorded total water resources, and Z value was selected. The difference between the water yield obtained by using this value and the total water resources was the smallest. We had added the manuscript in the line 657-661.

Point 13Questions about capitalization, grammar, language expression, etc.

Response 13: This part is revised one by one according to your suggestions. For example, "was" is changed to "were", left a space between the number and a dash.

Point 14It would be better to incorporate the Section "3. Results" and the Section "4. Discussion" in one section: "3. Results and Discussion".

Response 14: According to reviewer’ comments, we have added "3. Results" and "4.  Discussion" into "3. Results and Discussion".

Point 15The English language in the text is not good and it refers not only to the parts marked in the text but to the entire text. Therefore, it is necessary that the entire text be checked and corrected by a native English speaker or someone with good proficiency in English.

Response 15: According to reviewer’ comments, we have used English editing services to improve the level of English expression and corrected according to the suggestions.

Round 2

Author Response

Dear Reviewer,

Thank you for your helpful comments on our manuscript again. The main corrections in the paper and the response to the comments are as following.

Major Comments:

Point 1You have substantially upgraded your introduction but need to improve the connectivity in some parts. For instance, the connection between the 6th (lines 75 to 90) and 7th (91-98) paragraphs should be improved (Introduction section).

Response 1: According to reviewer’ comments, we added the content between the 6th (lines 75 to 90) and 7th (91-98) paragraphs: "Land use affects regional water yield by changing runoff and infiltration."

Point 2The physical processes that should be considered (together with the InVEST model) are those related to water yields, such as soil hydraulic conductivity, groundwater storage and release, vegetation type, root depth, … Then, you can explain how the InVEST model works and how of these processes are considered or not. You have only considered the InVEST model operation in the lines 55-58.

Response 2: According to reviewer’ comments, the work of the InVEST model is as follows: The water yield of InVEST model comes from precipitation minus actual evapotranspiration. In the aboveground, the actual evapotranspiration comes from the vegetation transpiration and surface evaporation, and the vegetation transpiration is related to the vegetation coefficient, seasonal constant and other parameters; in the underground and soil, the water is discharged through the root system, and then further evaporated by the plant, which is reflected in the parameters such as the effective available water for vegetation, the available water for vegetation, and the root depth in the model. Meanwhile, we have added these contents to lines 59-65 of the paper.

Point 3"For example, we can increase regional water yield by increasing vegetation coverage, reducing bare land area, reducing evapotranspiration, and improving water conservation capacity." You answered the question, but we need a reference here.

Response 3: According to reviewer’ comments, we added an article by Yang et al here.

Yang, X.; Chen, R.; Michael E.M.; Ji, G.; Xu, J.H. Modelling water yield with the InVEST model in a data scarce region of northwest China. Water Supply 2020, 20, 1035–1045.

Point 4You need to provide references to support your statements:

"Through the concept of "sponge city" proposed previously, the built up land can increase the water conservation capacity of urban construction area".

"Among the cultivated land, forest land, grassland, and build up land, the water conservation capacity of build up land is the strongest, and the water infiltration can be changed through the transformation of cultivated land and grassland to build up land."

Response 4: According to reviewer’ comments, we added an article by Zhang et al in the first part, and added an article by Dou et al in the second part.

Zhang,Y. The concept and function of "sponge city". Soil and Water Conservation Science and Technology in Shanxi 2021, 1, 1-2+21.

Dou, P.; Zuo, S.; Ren, Y. The impacts of climate and land use / land cover changes on water yield service in Ningbo region. Acta Scientiae Circumstantiae, 2019, 39, 2398-2409.

Point 5Please, just add the information about the stations being displayed in Figure 1. It can be like "we selected 46 stations (Figure 1) with complete.. ".

Response 5: According to reviewer’ comments, we added "(Figure 1)" in the line 243. Modified as "we selected 46 stations (Figure 1) with complete daily precipitation data from 1961 to 2005 from all meteorological stations in Shaanxi Province".

Point 6All models have their assumptions, uncertainty, and limitations. These must be highlighted in the work as you have done so far. Future studies should address these limitations to improve the water yield / watershed conservation conditions.

Response 6: According to reviewer’ comments, the assumptions, uncertainties and limitations of the InVEST model were shown in the discussion section "3.5.2. Limitations of water yield assessment and 3.5.3. Reflections on the InVEST Model".

Point 7Soil seems to be an important factor for water yield spatial distribution in your region. But, why? You should discuss a little about soil characteristics that are likely increasing and decreasing water yield. Please add this discussion to the final version.

Response 7: According to reviewer’ comments, we added the following content: "In each land use type in Shaanxi Province (Figure 3), the water yield depth of ferruginous soil, eluvial soil and alpine soil is 445.34 mm, 365.39 mm and 352.86 mm respectively, which is the soil type with high water yield distribution; the water yield depth of saline-alkali soil, hydrous soil and arid soil is 55.93 mm, 25.75 mm and 13.47 mm respectively, which is the soil type with low water yield distribution"

Minor comments:

Point 1I do not agree. Looking at Figure 4, you realize that the water yield was much more sensible to precipitation variability than actual evapotranspiration. Why did you describe in the text only the "strong" correlation between evapotranspiration and precipitation? Water yield was not more influenced (see how the water yield line follows precipitation variability)? How the changes in precipitation was opposite to that of water yield (your answer)? I could not see that in Figure 4. You must either add the importance of precipitation variability to water yield (because it basically drove its behavior) or correct Figure 4.

Response 1: According to reviewer’ comments, after the relationship between precipitation and evapotranspiration, we added the following: "Compared with actual evapotranspiration and potential evapotranspiration, precipitation has the strongest correlation with water yield. In the InVEST model, the water yield follows the principle of water balance. The regional water yield is precipitation minus actual evapotranspiration, but the actual evapotranspiration is also affected by precipitation, so precipitation plays a major role in the water yield".

Point 2Yes, your answer is right and what I have expected. But I could not find it in the text (Discussion section). Also, do you think that a "reduction of evaporation from groundwater" is what happens? Evaporation occurs in the soil surface (not in groundwater). Groundwater can be withdrawn by the root system to be further transpired by plants. Please correct this and focuses your discussion on the improved soil hydraulic characteristics (as you mentioned in your answer).

Response 2: According to reviewer’ comments, I have placed the contents explained in lines 647-654 of the paper. I'm very sorry that I mistakenly wrote "surface water" as "groundwater". Thank you very much for your reminding. At present, I have made corrections to it.

Point 3It sounds weird. I could not distinguish what "actual measured water resources" is. Does it mean all measured hydrological variables (e.g., soil moisture, evapotranspiration, streamflow, rainfall, snowfall, …)? I suggest you replace this expression.

Response 3: According to reviewer’ comments, in the paper, we want to express the total water resources of Shaanxi Province recorded in the Water Resources Bulletin. We have changed "measured water resources" to "recorded water resources".

Reviewer 3 Report

I am approaching my observations and corrections:  

Comments

   The work has been improved, because some corrections have been made, but some suggestions have remained uncorrected;

1.     The explanation of the abbreviations RCP2.6, RCP4.5, and RCP8.5 should be added in the Abstract;

2.     The labels of the numbers of the equations should be placed to the right of the equations and be aligned;

3.     An empty line should be left between the text and the equations, also between the Figures and the text and between the tables and the text;

Please find the additional comments in the attachment.

Author Response

Dear Reviewer,

Thank you for your helpful comments on our manuscript again. The main corrections in the paper and the response to the comments are as following.

Point 1The explanation of the abbreviations RCP2.6, RCP4.5, and RCP8.5 should be added in the Abstract.

Response 1: "RCP is the abbreviation of "Representative Concentration Pathways", which measures the concentration of greenhouse gases in the atmosphere. RCP2.6 represents the path with the lowest greenhouse gas emissions, followed by RCP4.5, and RCP8.5 represents the path with the highest greenhouse gas emissions." These contents have been added to "2.3.2 SDSM Model" and the full name of RCP "Representative Concentration Pathways" has been added to the abstract.

Point 2The labels of the numbers of the equations should be placed to the right of the equations and be aligned.

Response 2: According to reviewer’ comments, we have placed the labels of the numbers of the equations on the right side of the equations and used the right-aligned format for the labels of the numbers of the equations.

Point 3An empty line should be left between the text and the equations, also between the Figures and the text and between the tables and the text.

Response 3: According to reviewer’ comments, we have left an empty line between the text and the equations, also between the Figures and the text and between the tables and the text.
